# Temperature-dependent aqueous OH kinetics of C$_2$-C$_{10}$ linear and terpenoid alcohols and diols: new rate coefficients, structure-activity relationship and atmospheric lifetimes

Bartłomiej Witkowski,[1*] Priyanka Jain,[1] Beata Wileńska,[1] and Tomasz Gierczak[1]

[1]University of Warsaw, Faculty of Chemistry, al. Żwirki i Wigury 101, 02-089 Warsaw, Poland

Correspondence to: Bartłomiej Witkowski (bwitk@chem.uw.edu.pl)

**Abstract.** Aliphatic alcohols (AAs), including terpenoic alcohols (TAs), are ubiquitous in the atmosphere due to their widespread emissions from natural and man-made sources. Hydroxyl radical (OH) is the most important atmospheric oxidant in both aqueous and gas phases. Consequently, the aqueous oxidation of the TAs by the OH inside clouds and fogs is a potential source of aqueous secondary organic aerosols ($_{aq}$SOAs). However, the kinetic data, necessary for estimating the time scales of such reactions, remains limited. Here, bimolecular rate coefficients ($k_{OH_{aq}}$) for the aqueous oxidation of twenty-nine, C$_2$-C$_{10}$ AAs by hydroxyl radicals (OH) were measured with the relative rate technique in the temperature range 278 - 328 K. The values of $k_{OH_{aq}}$ for the fifteen AAs studied in this work were measured for the first time after validating the experimental approach. The $k_{OH_{aq}}$ values measured for the C$_2$-C$_{10}$ AAs at 298K ranged from $1.80\times10^9$ to $6.5\times10^9$ M$^{-1}$s$^{-1}$. The values of activation parameters, activation energy (7-17 kJ/mol), and average Gibbs free energy of activation (18±2 kJ/mol), strongly indicated the predominance of the H-atom abstraction mechanism. The estimated rates of the complete diffusion-limited reactions revealed up to 44% diffusion contribution for the C$_8$-C$_{10}$ AAs.

The data acquired in this work, and the values of $k_{OH_{aq}}$ for AAs, carboxylic acids, and carboxylate ions available in the literature, were used to develop a modified structure-activity relationship (SAR). The SAR optimized in this work estimated the temperature-dependent $k_{OH_{aq}}$ for all compounds under investigation with much higher accuracy as compared with the previous models. In the new model, an additional neighboring parameter was introduced (F≥(CH$_2$)$_6$, using the $k_{OH_{aq}}$ values for the homolog (C$_2$-C$_{10}$) linear alcohols and diols. A good overall accuracy of the new SAR at 298K (slope=1.022, R$^2$=0.855) was obtained for the AAs and carboxylic acids under investigation. The kinetic database ($k_{OH_{aq}}$ values in this work and compiled literature data) was also used to further

enhance the ability of SAR to predict temperature-dependent values of $k_{OH_{aq}}$ in the temperature range 278 - 328 K.

The calculated atmospheric lifetimes indicate that terpenoic alcohols and diols can react with the OH in aerosol, cloud, and fog water with (LWC≥0.1 g/m$^3$) and (LWC≥10$^{-4}$ g/m$^3$), respectively. The preference of terpenoic diols to undergo aqueous oxidation by the OH under realistic atmospheric conditions is comparable with terpenoic acids, making them potentially effective precursors of $_{aq}$SOAs. In clouds, a decrease in the temperature will strongly promote the aqueous reaction with the OH, primarily due to the increased partitioning of WSOCs into the aqueous phase.

## 1 Introduction

Biogenic volatile organic compounds (BVOCs) account for 20-90% of the global emissions of non-methane organics into the atmosphere, consisting primarily of isoprene and terpenes, which are emitted by vegetation (Sindelarova et al., 2022). The oxidation of these atmospherically abundant BVOCs in the gas phase is one of the major sources of secondary organic aerosols (SOAs), which are formed by the nucleation of the low-volatility, oxygenated products and their condensation onto the existing particles (Hallquist et al., 2009).

SOAs account for a large fraction of organic aerosol (OA) mass, which contributes up to 90% of fine (submicron) particulate matter (PM) (Jimenez et al., 2009; Xu et al., 2021). SOAs are important climate-forcing agents; they alter the properties of clouds, which in turn affects the Earth's hydrological cycle (Shrivastava et al., 2017; Mahilang et al., 2021). Furthermore, SOAs absorb and scatter solar radiation, thereby affecting the Earth's radiation balance and visibility (Tsigaridis and Kanakidou, 2018). Additionally, inhalation of fine PM has been associated with cardiovascular and respiratory conditions, resulting in increased mortality (Pye et al., 2021).

There are still many uncertainties regarding the formation, evolution, and climate forcing of SOAs, which greatly limits the current understanding of the present and future environmental impacts of aerosols (Fuzzi et al., 2015; Shrivastava et al., 2017; Mahilang et al., 2021). Historically, gas-phase oxidation of BVOCs was considered the major source of SOA in the atmosphere (Hallquist et al., 2009). More recently, the aqueous and multiphase reactions of atmospherically abundant water-soluble organic compounds (WSOCs) have been recognized as atmospherically relevant processes (Su et al., 2020; Carlton et al., 2020). These aqueous and multiphase reactions, although still not well characterized, are expected to significantly contribute to the formation of atmospheric SOAs (McNeill, 2015; McVay and Ervens, 2017; Ervens et al., 2018). For instance, the formation of SOAs in the

aqueous phase (referred to as $_{aq}$SOAs) can, in part, explain the discrepancies between observed and modeled

budgets of OAs (Lin et al., 2014; Tsui et al., 2019; Su et al., 2020; Pai et al., 2020).

Alcohols are one of the major classes of WSOCs in the atmosphere; they are emitted from both natural and man-made sources (Mellouki et al., 2015), and include short and longer-chain (fatty) linear alcohols and diols, as well as cyclic and terpenoic alcohols (TAs) (Guenther et al., 2012; Chen et al., 2021; Konjević et al., 2023). In the atmosphere, hydroxyl radical (OH) is the most important daytime oxidant (Herrmann et al., 2015; Mahilang et al.,

2021). Consequently, the aliphatic WSOCs, including AAs, are expected to react primarily with OH and $NO_3$ (Mellouki et al., 2015). In addition to reaction in the gas phase (Ceacero-Vega et al., 2012), AAs can also undergo aqueous oxidation by the OH - R(I) - inside the aqueous particles, depending on their Henry's law constants (H, M/atm$^{-1}$) and liquid water content (LWC, g/m$^3$) of clouds, fogs, and aerosols (Sander, 2015; Herrmann et al., 2015).

$$AA + OH_{aq} \rightarrow products \quad R(I)$$

Consequently, in the atmospheric aqueous droplets, R(I) can yield low-volatility products, including carboxylic (terpenoic) acids (Yasmeen et al., 2011; Witkowski et al., 2023). Therefore, TAs (and diols) may be efficient precursors of $_{aq}$SOAs. Due to their larger carbon backbones ($C_7$-$C_{10}$) and cyclic structures, TAs have the potential to undergo R(I) without extensive fragmentation (Ceacero-Vega et al., 2012). However, to date, there have not

been many studies that investigated the aqueous OH kinetics of TAs (Hoffmann et al., 2009; Gligorovski et al., 2009; Herrmann et al., 2015; Leviss et al., 2016).

To evaluate whether or not a reaction with the OH is a relevant removal pathway for a given molecule under realistic atmospheric conditions, chemical models are utilized (Bräuer et al., 2019; Zhu et al., 2020). Bimolecular reaction rate coefficients for the reaction of organic compounds with the OH in the gas ($k_{OH_{gas}}$) and in the aqueous

($k_{OH_{aq}}$) phases are the foundation of both simple and complex chemical models (Herrmann et al., 2015), used in the field of atmospheric chemistry to investigate the reaction pathways for various WSOCs (Zhu et al., 2020).

Following decades of extensive research, gas-kinetic databases are available for the reactions of various VOCs with atmospherically abundant oxidants (McGillen et al., 2020; McGillen et al., 2021). On the other hand, the currently available aqueous kinetic datasets are still limited and contain primarily room-temperature data (Buxton

et al., 1988; Herrmann et al., 2015). Reliable kinetic datasets are crucial for an accurate parametrization of the complex aqueous and multiphase reactions, which are directly connected with the formation and processing of $_{aq}$SOAs in the atmosphere (Ervens, 2015; Bräuer et al., 2019).

In this work, after validating the experimental approach, the values of $k_{OH_{aq}}$ for linear and terpenoic AAs and diols were measured with the relative rate technique. Here, $k_{OH_{aq}}$ for multiple AAs were measured in a single experiment; R(I) was carried out in a bulk reactor and the concentrations of individual AAs in the reaction solution were measured with gas chromatography (Herrmann et al., 2005). Because the temperature range in the atmosphere extends above and below room temperature, the kinetic measurements were carried out in the temperature range between 278 and 323 K.

However, measuring the $k_{OH}$ values for the numerous WSOCs present in the environment is unfeasible (Goldstein and Galbally, 2007). Consequently, structure-activity relationships (SARs) are widely utilized in the field of atmospheric chemistry for estimating the $k_{OH}$ values of various organic molecules based on their molecular structures (Kwok and Atkinson, 1995; Monod and Doussin, 2008). At the same time, the ability of current aqueous SARs to predict the temperature-dependent $k_{OH_{aq}}$ values for the WSOCs with larger carbon backbones remain limited (Herrmann et al., 2005; Bräuer et al., 2019). Moreover, the methodology used in the current aqueous SARs was developed for predicting the values of $k_{OH_{gas}}$. Therefore, it is still unclear whether or not this approach is entirely adequate for estimating the rates of aqueous reactions (Kwok and Atkinson, 1995; Monod and Doussin, 2008).

The kinetic data acquired in this work, and the compiled literature data, were used to develop a modified kinetic SAR for predicting the values of $k_{OH_{aq}}$ for AAs, carboxylic acids, and carboxylate ions at different temperatures. Subsequently, the atmospheric lifetimes for the five TAs under investigation due to R(I) were estimated to evaluate their potential to yield $_{aq}$SOAs under realistic atmospheric conditions.

## 2 Materials and methods

Materials and reagents are listed in section S1 in the Electronic Supplementary Material (ESM). Deionized (DI) water ($\geq$18 M$\Omega\times$cm$^{-1}$) was used in all experiments.

### 2.1 Aqueous photoreactor

The photoreactor is described in more detail in our previous study (Witkowski et al., 2019). The reaction vessel was a jacketed, quartz flask with an internal volume of 0.1 L. Radicals (OH) were generated *in situ* by photolyzing hydrogen peroxide (H$_2$O$_2$). The reaction vessel was surrounded by eight lamps (TUV TL 4W, Philips, $\lambda_{max}$=254 nm). In some experiments, an ozone-free, 2.12" pen-ray lamp (11SC-2.12-PO-16-800, Analytik Jena, $\lambda_{max}$=254

110    nm) was used. The pen-ray lamp was immersed in the reaction solution inside a quartz vial. Temperatures of the reaction solution were adjusted with a circulating water bath (SC100-A10, Thermo Fisher Scientific). The solution temperature was additionally monitored with an externally calibrated sensor (TP-361, Czaki Thermo-Product).

## 2.2 Chromatographic analyses

Semi-quantitative analyses of AAs were carried out with gas chromatography (GC) coupled with the flame
ionization detector (FID) or with mass spectrometry (MS).

GC/MS analyses were carried out using a GC/MS-QP2010 Ultra (Shimadzu) gas chromatograph coupled with a single-quadrupole mass spectrometer equipped with an electron ionization (70 eV) ion source. The instrument was equipped with an AOC-5000 autosampler (Shimadzu). Analytes were separated with a VF-WAXms column (Agilent) or with a ZB-5MSplus column (Phenomenex) - section S2.1. The mass spectrometer was operating in
the selected ion monitoring mode (Table S1 and S2).

GC/FID analyses were carried out using a GC17A gas chromatograph (Shimadzu). Analytes were separated with a ZB-Waxplus capillary column (Phenomenex). GC/FID instrument was used for the analysis of $C_5$-$C_{10}$ AAs, as well as for the $C_2$-$C_4$ linear AAs (Tables S3 and S4).

## 2.3 Experimental procedures

AAs under investigation were separated into four groups, based on their OH reactivities, chromatographic peak shape on a given stationary phase, GC detector response, and recoveries obtained following liquid-liquid extraction of aqueous samples taken from the photoreactor with ethyl acetate (Tables S1 – S4).

The concentration of cyclic, terpenoic, and $C_5$-$C_{10}$ AAs (Tables S1, S2, and S4) in the reaction solution was 0.2 - 0.4 mM each, and the concentration of $H_2O_2$ was 0.1 M. The concentration of $C_2$-$C_4$ AAs (Table S3) in the reaction
solution was 0.8 – 1.0 mM each, and the concentration of $H_2O_2$ in these experiments was 0.4 M.

AAs under investigation were first dissolved in water and, the resulting solution was filtered through a 0.7 μm GF syringe filter and transferred to the reaction vessel. The temperature of the solution was allowed to stabilize before adding $H_2O_2$, which initiated the reaction.

Thirteen aliquots were sampled from the reaction solution and the reaction time was 15-70 mins, depending on
the temperature and concentration of $H_2O_2$. The $C_2$-$C_4$ AAs (Table S3) were analyzed by injecting 0.5 μl of the aqueous sample taken from the reactor into the GC/FID instrument. For the rest of the AAs under investigation, 800 μl of the reaction solution was saturated with NaCl, and extracted with 300 μl of ethyl acetate containing 0.3

mM of dimethyl phthalate (internal standard). Afterward, the organic layer was dried with anhydrous sodium sulfate ($Na_2SO_4$) and analyzed with GC/MS or GC/FID instrument.

## 2.4 Relative rate technique and activation parameters

The $k_{OH_{aq}}$ values for the AAs under investigation were derived with eq. I.

$$Ln\left(\frac{[AA]_0}{[AA]_t}\right) = \frac{k_{AA}}{k_{Ref}} Ln\left(\frac{[Ref]_0}{[Ref]_t}\right) \quad (I)$$

In eq. I, [AA], and [Ref] are the initial (0) and intermediate concentrations of the AAs under investigation and the kinetic reference compound, respectively. $k_{Ref}$ and $k_{AA}$ ($M^{-1}s^{-1}$) are the $k_{OH_{aq}}$ for the kinetic reference compound and the AAs under investigation at a given temperature, respectively. 1,4-Butanediol and 1,5-pentanediol: $k_{OH_{aq}}$ at 298K = $(3.5\pm0.1)\times10^9\,M^{-1}s^{-1}$ and $(4.4\pm0.7)\times10^9\,M^{-1}s^{-1}$, respectively, were used as the primary kinetic reference compounds (Hoffmann et al., 2009).

The temperature-dependent $k_{OH_{aq}}$ values were used to obtain the pre-exponential factors (A) and activation energies ($E_a$) using the Arrhenius expression (eq. II).

$$Ln(k_{OH_{aq}}) = Ln(A) - \left(\frac{E_a}{R}\right) \cdot \frac{1}{T} \quad (II)$$

In eq. II, $k_{OH_{aq}}$ is the bimolecular reaction rate coefficient of the reaction of AAs with the OH ($M^{-1}\,s^{-1}$) at a given temperature, T is the temperature (K), R is the gas constant ($kJ\,K^{-1}\,mol^{-1}$), A is the pre-exponential factor ($M^{-1}\,s^{-1}$) and $E_a$ is the activation energy ($kJ\,mol^{-1}$). The values of Gibbs free energy of activation ($\Delta G^{\ddagger}$), the enthalpy of activation ($\Delta H^{\ddagger}$), and the entropy of activation ($\Delta S^{\ddagger}$) were also calculated for each AA, using the temperature-dependent $k_{OH_{aq}}$ values (section S4). The rates of diffusion-limited reactions ($k_{diff}$, $M^{-1}s^{-1}$) were calculated with the Smoluchowski equation (section S5) (Schöne et al., 2014; Kroflič et al., 2020).

## 2.5 Structure-activity relationship

The SAR methodology used in this work was introduced by (Atkinson, 1987), and later used to develop SAR for estimating the values of $k_{OH_{aq}}$ of aliphatic molecules containing C, H, and O atoms (Monod and Doussin, 2008; Doussin and Monod, 2013). SAR parameters include F and G substituent factors for $CH_3$, $CH_2$, CH, C, OH, FC=O, COOH, COO- moieties, F-factors for the $C_6$, $C_5$, $C_4$ rings, and base rate coefficients for the H-atom abstraction from $CH_3$, $CH_2$, CH and -OH groups (Fig. 1).

A)

1) $k(CH) \times F(OH)_9 \times F(C)_{10} \times F(C)_{11} \times G(CH_3)_2 \times G(CH_3)_3 \times G(CH_3)_7 \times G(CH)_4 \times G(CH_2)_6 \times G(CH_2)_8 \times F(C_6) \times F(C_5)$
2) $k(CH_3) \times F(C)_{10} \times G(CH_3)_3 \times G(CH)_4 \times G(CH)_1$
3) $k(CH_3) \times F(C)_{10} \times G(CH_3)_2 \times G(CH)_4 \times G(CH)_1$
4) $k(CH) \times F(CH_2)_5 \times F(CH_2)_8 \times F(C)_{10} \times G(CH_2)_6 \times G(C)_{11} \times G(CH)_1 \times G(CH_3)_2 \times G(CH_3)_3 \times F(C_6) \times F(C_5)$
5) $k(CH_2) \times F(CH_2)_6 \times F(CH)_4 \times G(C)_{10} \times G(CH_2)_8 \times G(C)_{11} \times F(C_6) \times F(C_5)$
6) $k(CH_2) \times F(C)_{11} \times F(CH_2)_5 \times G(CH_3)_7 \times G(CH_2)_8 \times G(CH)_1 \times G(CH)_4 \times F(C_6) \times F(C_5)$
7) $k(CH_3) \times F(C)_{11} \times G(CH)_1 \times G(CH_2)_6 \times G(CH_2)_8$
8) $k(CH_2) \times F(CH)_4 \times F(C)_{11} \times G(C)_{10} \times G(CH_2)_5 \times G(CH)_1 \times G(CH_3)_7 \times G(CH_2)_6 \times F(C_5) \times F(C_5)$
9) $k(OH) \times F(CH)_1 \times G(C)_{10} \times G(C)_{11}$

$$k_{OH}(\text{Fenchol}) = \sum_{n=1}^{9} (k_n(\text{adjusted partial rate coefficients}))$$

B)

1) $k(CH_2) \times F(OH)_8 \times F(CH_2)_2 \times G(CH_3)_3 \times F((CH_2)_6)$
2) $k(CH_2) \times F(CH_2)_1 \times F(CH_2)_3 \times G(OH)_8 \times G(CH_2)_4 \times F((CH_2)_6)$
3) $k(CH_2) \times F(CH_2)_2 \times F(CH_2)_4 \times G(CH_2)_1 \times G(CH_2)_5 \times F((CH_2)_6)$
4) $k(CH_2) \times F(CH_2)_5 \times F(CH_2)_3 \times G(CH_2)_2 \times G(CH_2)_6 \times F((CH_2)_6)$
5) $k(CH_2) \times F(CH_2)_6 \times F(CH_2)_4 \times G(CH_2)_7 \times G(CH_2)_3 \times F((CH_2)_6)$
6) $k(CH_2) \times F(CH_2)_5 \times F(CH_2)_7 \times G(CH_2)_4 \times F((CH_2)_6)$
7) $k(CH_3) \times F(CH_2)_6 \times G(CH_2)_5 \times F((CH_2)_6)$
8) $k(OH) \times F(CH_2)_1 \times G(CH_2)_2$

$$k_{OH}(\text{1-Heptanol}) = \sum_{n=1}^{8} (k_n(\text{adjusted partial rate coefficients}))$$

**Figure 1: Examples of estimating the values of $k_{OH_{aq}}$ with SAR for fenchol (A) and 1-heptanol (B).**

The partial rate coefficients are modified by the functional groups in the α and β-positions using the F and G substituent factors (Monod and Doussin, 2008; Doussin and Monod, 2013). Note that fenchol has two fused $C_5$ rings and one $C_6$ ring, and the partial k values for H-atom abstraction connected with these rings are adjusted with the $F(C_5)$, and $F(C_6)$ factors - Fig. 1A (Monod and Doussin, 2008). Furthermore, in this work, an additional F

factor was introduced (section 3.2) for the H-atoms connected with a straight-chain carbon backbone$\geq C_6$ (Fig. 1B).

A dataset, containing the values of $k_{OH_{aq}}$ (298 K) for 56 AAs, carboxylic acids, and carboxylate anions (Witkowski, 2023), was used to derive the values of F and G factors by minimizing values of Q parameter (eq. III) using the Excel Solver routine (González-Sánchez et al., 2021).

$$Q = \sum_n \frac{(k_{SAR} - k_{exp})^2}{\sigma_n^2} \qquad \text{(III)}$$

In eq. III, $k_{SAR}$, and $k_{exp}$ are estimated and measured $k_{OH_{aq}}$ values at 298K and σ is the experimental uncertainty. The temperature dependence of the partial $k_{OH}$ values for $CH_3$, $CH_2$, CH, and OH groups is described via eq. IV using C and D factors. (Kwok and Atkinson, 1995) C and D factors in eq. IV were adjusted using a database of 351 temperature-dependent rate coefficients (Witkowski, 2023).

$$k_{OH_{partial}}(T) = CT^2 \times exp\left(\frac{-D}{T}\right) \quad \text{(IV)}$$

In eq. IV, T is the reaction temperature (K). Furthermore, the values of F and G factors were adjusted at different temperatures with eq. V.

$$F \text{ or } G(T) = exp\left(\frac{298 \times ln(F \text{ or } G \text{ at } 298K)}{T}\right) \qquad \text{(V)}$$

After adjusting the values of the neighboring parameters (F and G), they were kept unchanged and the values of

D and C factors (eq. IV) were adjusted with the measured, temperature-dependent $k_{OH_{aq}}$ values for AAs (this work) and for carboxylic acids, and carboxylate (di)anions (Witkowski, 2023).

**2.6 Control measurements and uncertainty**

The reported uncertainties of the $k_{OH_{aq}}$ values measured in this work were derived with the exact differential method (eq. VI), taking into account the uncertainties of the slopes of kinetic plots (eq. I) and the uncertainties of

the $k_{ref}$ values reported in the literature (Schöne et al., 2014) – see section S6 for a detailed derivation of this formula.

$$\Delta k_{OH_{aq}} = \sqrt{\left(slope \times \Delta k_{ref}\right)^2 + \left(k_{ref} \times \Delta slope\right)^2} \qquad \text{(VI)}$$

In eq.VI, the slope is the slope of the kinetic plot corresponding to the $k_{AA}/k_{ref}$ ratio (eq. I), and, $\Delta the$ slope is the experimental uncertainty, which was derived as 2σ from three or more separate measurements, $k_{ref}$ and $\Delta k_{ref}$

are the $k_{OH_{aq}}$ values for the kinetic reference compounds and their uncertainties, respectively (Hoffmann et al., 2009). Uncertainties of the $E_a$ values are reported as standard errors of the linear fitting (eq. II) and the

uncertainties for the other activation parameters (eq. SI-SIII) were calculated with the exact differential method. Control experiments carried out without turning on the lamp or without adding $H_2O_2$ to the reaction solution, confirmed that the AAs under investigation did not undergo direct photolysis or dark reactions with the $H_2O_2$, within the time scale of the experiments. To further confirm that the five terpenoic alcohols and diols under investigation were photochemically stable under the experimental conditions used, their UV-Vis spectra were acquired (Fig. S5). Also, no evaporation of the analytes was observed within the studied temperature range (section 2.1). Control experiments also confirmed that the AAs that were analyzed in the same batch (section S3) were not converted into one another during R(I).

## 3 Results and discussion

### 3.1 Results of the kinetic measurements

The unknown $k_{OH_{aq}}$ values were calculated using eq. I; for all experiments, the squared linear coefficients of determination ($R^2$)>0.99 were obtained, as shown in the representative relative kinetic plots (Fig. 2).

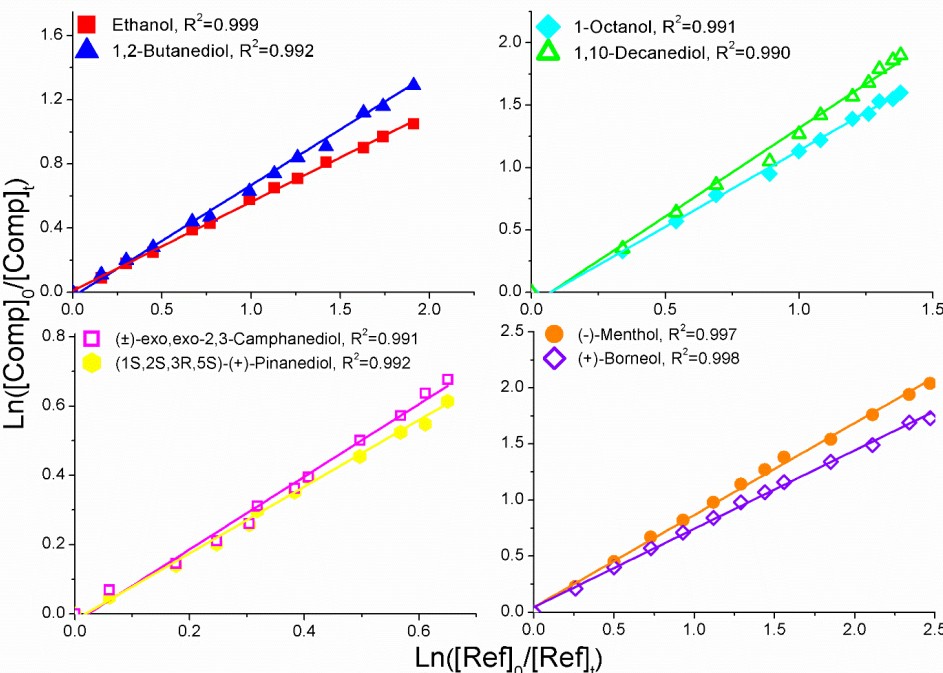

**Figure 2: Sample relative kinetic plots obtained in this work at 298K; data are represented by points, and lines are linear fits to the experimental data.**

The values of $k_{OH_{aq}}$ measured in the temperature range between 278 and 323K were used to calculate activation parameters. For all AAs under investigation, linear Arrhenius plots (eq. II) were obtained, as shown by the representative data presented in Fig. 3.

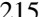215

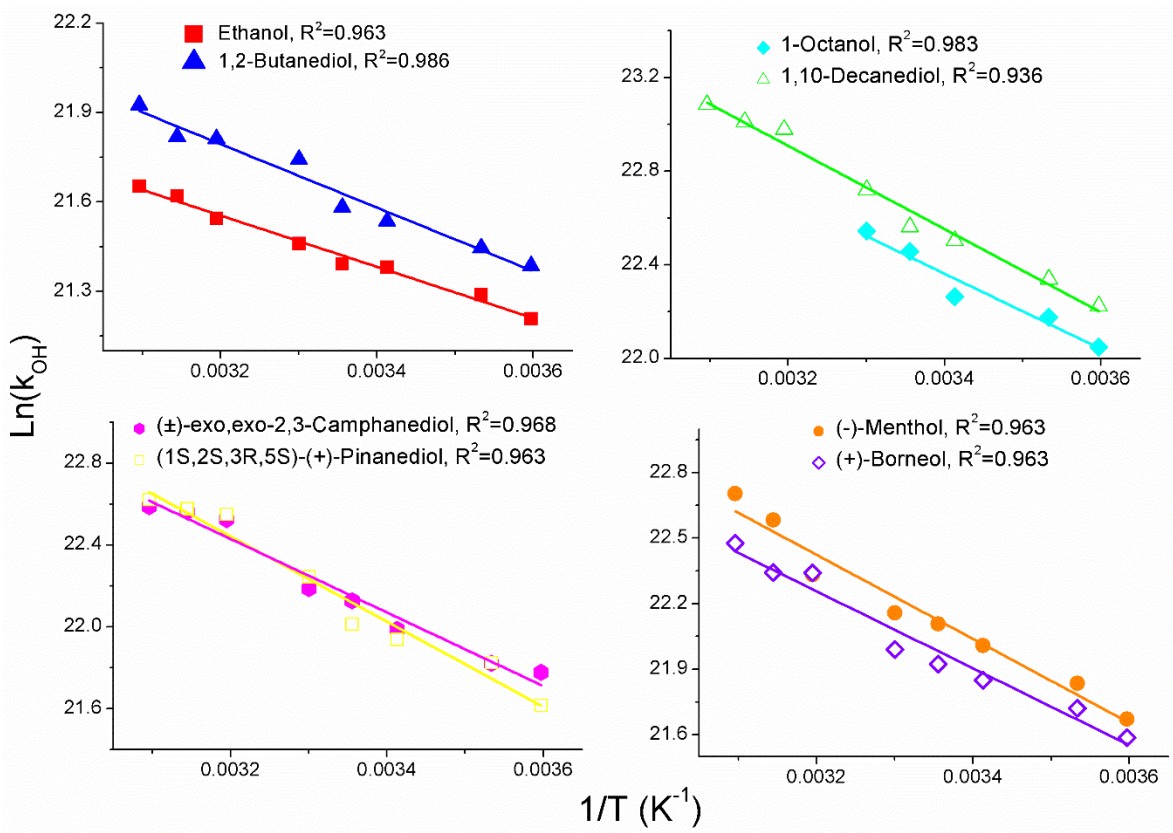

**Figure 3: Sample Arrhenius plots obtained in this work; data are represented by points, and lines are linear fits to the experimental data.**

The $k_{OH_{aq}}$ values measured at 298 K and the activation parameters for the AAs investigated in this work are listed

in Table 1; all temperature-dependent $k_{OH_{aq}}$ values measured in this work are listed in Table S6.

**Table 1.** The values of $k_{OH_{aq}}$ obtained in this work for the reaction of OH with AAs at 298K, literature data, and the preferred values

| Compound | $k_{OH}$ at 298 K, $(M^{-1}s^{-1}) \times 10^{-9}$ | | Ref. |
| | This work | Literature[a] | |
|---|---|---|---|
| Ethanol[b] | 2.0±0.1 | 2.0±0.4 | (IUPAC, 2019) |
| 1-Propanol | 2.5±0.2 | 3.2±0.6 | (IUPAC, 2019) |
| | | 3.8±0.7 | (Monod et al., 2005) |
| | | 3.2±0.2 | (Ervens et al., 2003) |
| | | 3.0±0.5 | (Willson et al., 1971) |
| | | 3.15±0.8 | (Scholes and Willson, 1967) |
| | | 2.6 | (Adams et al., 1965) |
| 1-Butanol | 3.2±0.2 | 4.3±0.4 | (IUPAC, 2019) |
| | | 4.1±0.8 | (Herrmann, 2003) |
| | | 4.2±0.4 | (Monod et al., 2005) |
| | | 4.5±0.5 | (Willson et al., 1971) |
| | | 5.1±0.4 | (Prutz and Vogel, 1976) |
| | | 3.7±1 | (Scholes and Willson, 1967) |
| | | 3.9 | (Adams et al., 1965) |
| 2-Butanol | 2.5±0.2 | 3.2±0.3 | (IUPAC, 2019) |
| | | 3.5±0.4 | (Herrmann, 2003) |
| | | 2.8±0.7 | (Scholes and Willson, 1967) |
| | | 3.1 | (Adams et al., 1965) |
| | | 2.4±0.2 | (Anbar et al., 1966) |
| 1-Pentanol | 4.5±0.3 | 4.7±0.7 | (IUPAC, 2019) |
| | | 5.1±0.2 | (Stemmler and von Gunten, 2000) |
| | | 3.8 | (Reuvers et al., 1973) |
| | | 5.3±1 | (Scholes and Willson, 1967) |
| | | 4.8±0.5 | (Anbar et al., 1966) |
| 1-Hexanol[c] | 4.9±0.4 | 5.9±1.9 | (IUPAC, 2019) |
| | | 5.9±1.5 | (Scholes and Willson, 1967) |
| 1-Heptanol[c] | 5.0±0.4 | 6.1±2 | (IUPAC, 2019) |
| | | 6.1±1 | (Scholes and Willson, 1967) |
| 1-Octanol[c] | 5.7±0.4 | 6.5±2 | (IUPAC, 2019) |

| | | 6.5±2 | (Scholes and Willson, 1967) |
|---|---|---|---|
| 1-Nonanol | 5.4±0.4 | | |
| 1-Decanol | 6.2±0.5 | | |
| 3-Ethyl-3-pentanol | 2.5±0.3 | | |
| 1,2-Ethanediol | 1.9±0.2 | 1.7±0.3 | (IUPAC, 2019) |
| | | 1.6±0.03 | (Hoffmann et al., 2009) |
| | | 2.6±0.5 | (Matheson et al., 1973) |
| | | 1.62 | (Willson et al., 1971) |
| | | 1.8±0.5 | (Scholes and Willson, 1967) |
| | | 1.5±0.3 | (Anbar et al., 1966) |
| | | 1.5±0.2 | (Adams et al., 1965) |
| 1,2-Propanediol | 1.8±0.1 | 1.75±0.6 | (IUPAC, 2019) |
| | | 1.6±0.3 | (Hoffmann et al., 2009) |
| | | 1.7 | (Adams et al., 1965) |
| | | 1.7±0.4 | (Scholes and Willson, 1967) |
| 1,2-Butanediol[c] | 2.4±0.2 | 2.2±0.4 | (IUPAC, 2019) |
| | | 2.2±0.4 | (Hoffmann et al., 2009) |
| 1,6-Hexanediol[c] | 4.9±0.4 | 4.6±1.5 | (IUPAC, 2019) |
| | | 4.7 | (Anbar et al., 1966) |
| 1,7-Heptanediol | 5.4±0.4 | | |
| 1,8-Octanediol | 5.5±0.4 | | |
| 1,9-Nonanediol | 6.4±0.4 | | |
| 1,10-Decanediol | 6.3±.0.4 | | |
| Cyclohexanol | 3.6±0.3 | | |
| *trans*-1,2-Cyclohexanediol | 2.9±0.1 | | |
| *exo*-Norborneol | 1.9±0.1 | | |
| *cis*-2-Methylcyclohexanol | 4.8±0.5 | | |
| (+)-Fenchol | 3.0±0.2 | | |
| (+)-Borneol | 3.3±0.1 | | |
| (−)-Menthol | 4.0±0.1 | | |
| (±)-*exo,exo*-2,3-Camphanediol | 4.1±0.1 | | |
| (1S,2S,3R,5S)-(+)-Pinanediol | 3.6±0.1 | | |

[a]Preferred values are based on the data compiled by the IUPAC Task Group on Atmospheric Chemical Kinetic Data Evaluation. Note also that for relative measurements, a large number of the $k_{OH_{aq}}$ listed in the original studies reviewed by the Task Group were recalculated using the updated $k_{ref}$ values for the kinetic reference compounds. [b]For clarity, only the preferred value is given for ethanol, due to a very large number of measured $k_{OH_{aq}}$ values for these two AAs. Note that the preferred values given in the IUPAC datasheets and their uncertainties reflect both the distribution and number of the previously measured $k_{OH_{aq}}$ values.

The majority of the $k_{OH_{aq}}$ values measured in this are work in very good agreement with the literature data, within the uncertainties reported (Table 1), thereby confirming that the experimental approach used in this work is reliable. To date, only one measured $k_{OH_{aq}}$ value is available for 1-hexanol, 1-heptanol, 1-octanol, 1,2-butanediol, 1,6-hexanediol (Anbar et al., 1966; Scholes and Willson, 1967; Hoffmann et al., 2009) and, the preferred values are based only on these single measurements (Table 1). For this reason, uncertainties of 33% were recommended for the $k_{OH_{aq}}$ values for these five AAs (IUPAC, 2019). At the same time, the $k_{OH_{aq}}$ values measured in this work for 1-hexanol, 1-heptanol, 1-octanol, 1,2-butanediol, 1,6-hexanediol are in good agreement with the preferred values, within the reported (and recommended) uncertainties.

For 1-propanol and 1 and 2-butanols, the values of $k_{OH_{aq}}$ measured in this work are noticeably lower than the preferred values (Table 1). However, these differences are difficult to explain. In the case of ethanol, 1,2-propanediol, 1,2-ethanediol and1,2-butanediol, for which the values of $k_{OH_{aq}}$ were measured together with 1-propanol and 1 and 2-butanols (Table S2), using the same kinetic reference compound (1,4-butanediol), an excellent agreement with the previously measured values was obtained (Table 1 - see also Fig. 4). One possible explanation may be that product peaks overlapped with the chromatographic peaks corresponding to AAs under investigation, which would lower the rate of disappearance of 1-propanol and 1 and 2-butanols during R(I) obtained from the GC/FID measurements. This would result in the lower $k_{OH_{aq}}$ measured for these three AAs. To test that hypothesis, additional experiments were carried out, by adding only 1-propanol, 1 and 2-butanols, and ethanol to the reaction solution, to minimize the possible interferences from the products. The $k_{OH_{aq}}$ values obtained from these additional measurements for 1-propanol and 1 and 2-butanols were the same (within the obtained uncertainties – Table S8) as those reported in Table 1.

To our knowledge, for the rest of the AAs investigated in this work, including $C_9$-$C_{10}$ n-alcohols, $C_7$-$C_{10}$ α,ω-diols, and, cyclic AAs and diols, the $k_{OH_{aq}}$ values were measured for the first time. The data obtained for these compounds are discussed in more detail later in this section.

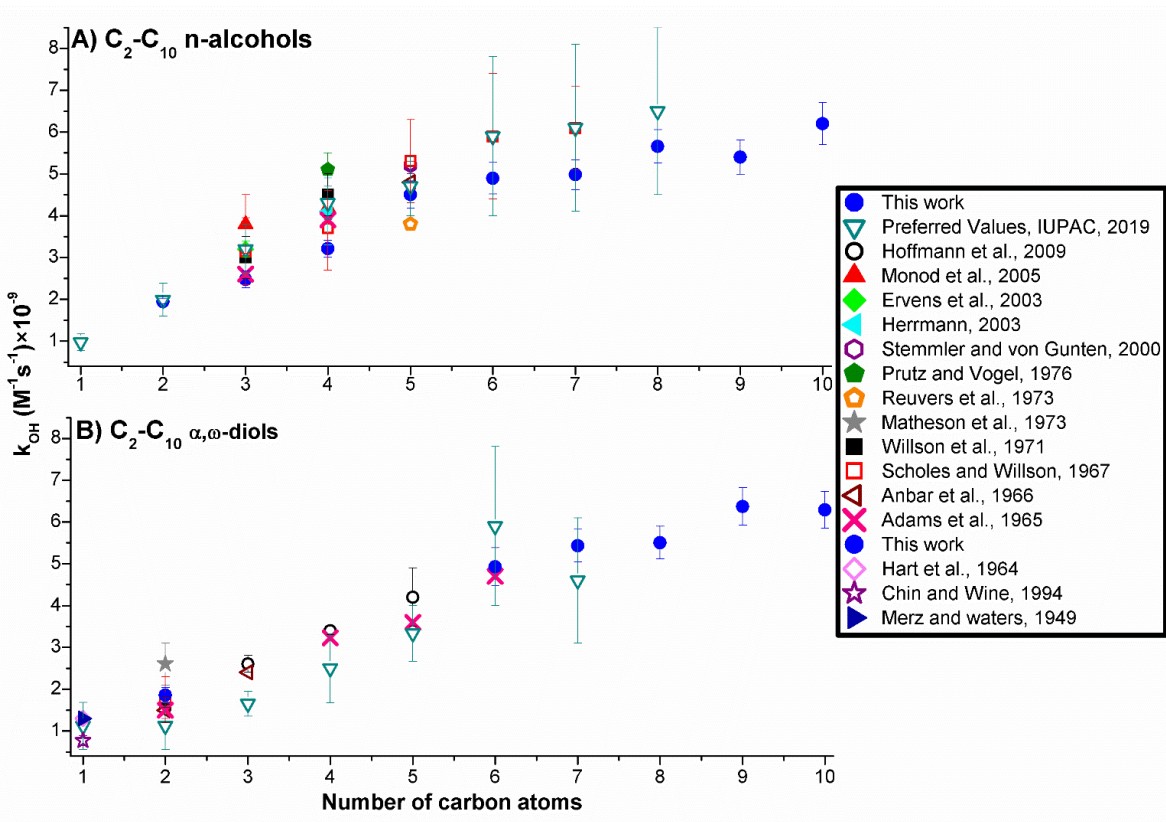

**Figure 4:** The $k_{OH_{aq}}$ values measured at 298K for the homolog series of n-alcohols (A) and α,ω-diols (B). For the $C_6$-$C_8$ n-alcohols and 1,6-hexanediol, the preferred values are based on a single, measured $k_{OH_{aq}}$ values, with a recommended uncertainty of 33%. For methanediol (formaldehyde hydrate), measured $k_{OH_{aq}}$ values from (Merz and Waters, 1947; Hart et al., 1964; Chin and Wine, 1992) are included.

The data acquired in this work, and the literature data, showed an incremental increase in the measured $k_{OH_{aq}}$ values in the homolog series of n-alcohols and α,ω-diols, which exhibited a distinctive curvature around four or five carbon atoms (Fig. 4A and B). In Fig. 5A, the preferred $k_{OH_{aq}}$ values (when available) for the AAs investigated in this work are superimposed with the data for n-alkanes. Very similar curvature was observed in the gas phase, for the homolog series of n-alcohols (Mellouki et al., 2003; Monod et al., 2005) – Fig. 5B.

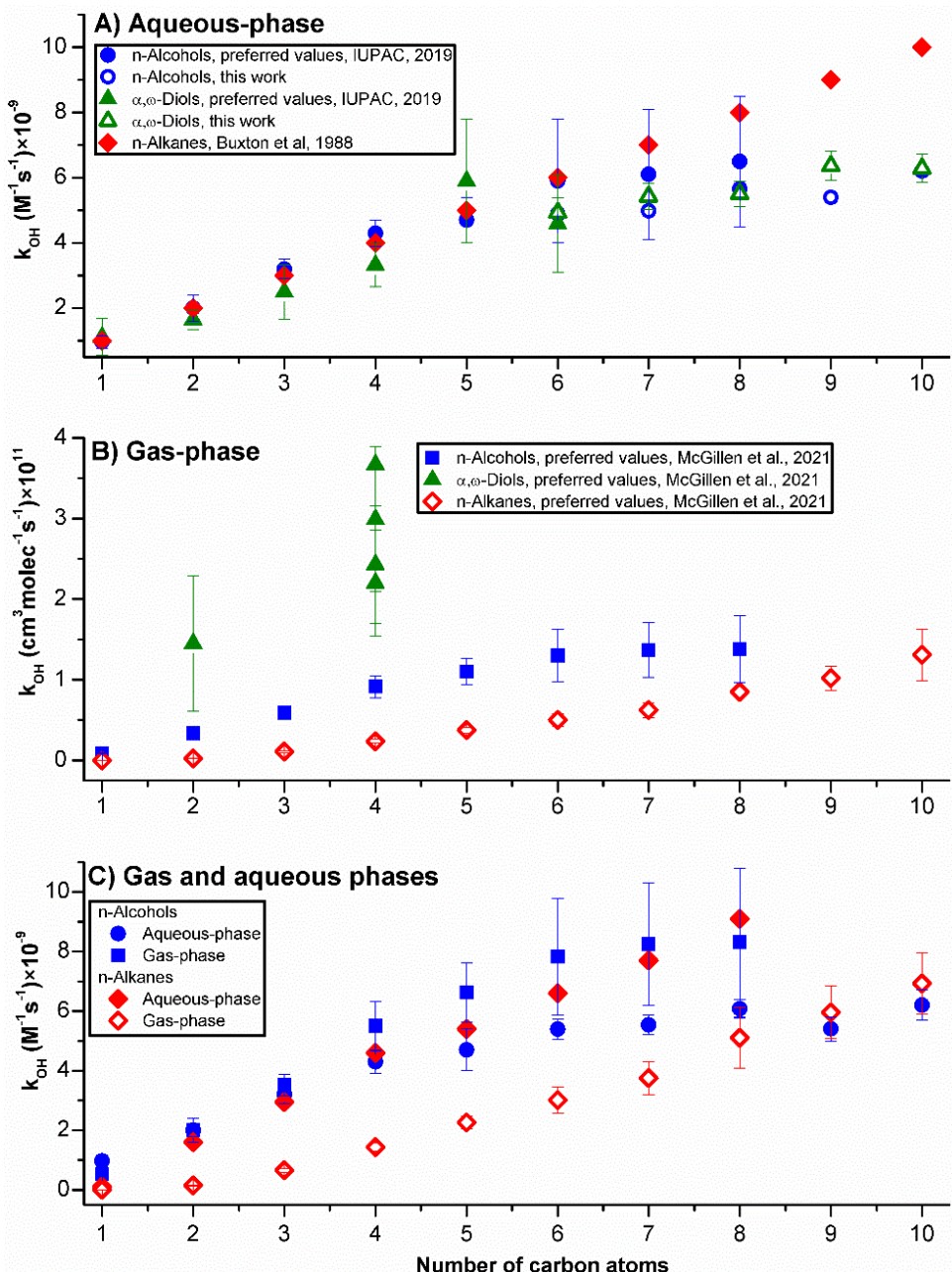

**Figure 5: Measured k$_{OH}$ values for n-alkanes, n-alcohols, and diols in the gas (A) and aqueous (B) phases. When available, preferred (recommended) values are shown for both phases (see also Table S5). (A) - for the 1-hexanol, 1-heptanol, 1-octanol, 1,2-butanediol, and 1,6-hexanediol, the preferred values are based on single measurements; hence for these AAs, the values of $k_{OH_{aq}}$ measured in this work are also included. (C) The preferred values are shown, when available and, diols were not included due to the very low number of $k_{OH_{aq}}$ values available to date. (C) For C$_6$-C$_{10}$ n-alcohols, average values (preferred values and data acquired in this work) are shown and, the uncertainties shown were obtained with the error propagation method. (A, C) No $k_{OH_{gas}}$ values are currently available for n-decanol and 1-nonanol. (A, C) For the C$_1$-C$_{10}$ n-alkanes, the majority of $k_{OH_{aq}}$ values are based on single measurements (Table S5) and, no preferred values (or their uncertainties) are currently available.**

In the gas phase, the curvature in Fig. 5B was attributed to the long-range activating effect of the -OH moieties, which extends to around four carbon atoms (Mellouki et al., 2003). This activating effect of -OH is due to the formation of a hydrogen-bonded (H-bonded) complex between OH and some oxygenated organics. The curving (Fig. 5B) is observed because the H-atom abstraction proceeds via a cyclic transition state (TS), which becomes less energetically favorable for the longer-chain molecules (Smith and Ravishankara, 2002). This mechanism is also confirmed by the limited data available for diols; the $k_{OH_{gas}}$ values for 1,2-ethanediol, 1,4-butanediol, 1,3-butanediol,1,2-butanediol, and 2,3-butanediol are significantly higher as compared with the corresponding n-alcohols and n-alkanes (Fig. 5B) (McGillen et al., 2021).

However, in the aqueous phase, the formation of this H-bonded complex is suppressed, most likely due to the H-bonds between the water molecules and the -OH moieties of AAs (Monod et al., 2005). The suppression of the formation of the H-bonded complex between OH and -OH moieties explains the lack of the activating effects of one or two -OH moieties in the aqueous phase; the values of $k_{OH_{aq}}$ are the same for $C_1$-$C_7$ n-alkanes, n-alcohols, and α,ω-diols (Fig. 5A). At the same time, the assumption that in the aqueous phase, n-alcohols, and α,ω-diols react via the same mechanism as n-alkanes (direct abstraction of aliphatic H-atoms by the OH) (Monod et al., 2005), does not explain the curvatures observed in Figs. 4A and B.

In the aqueous solutions, it was proposed that abstraction of aliphatic H-atoms by the OH proceeds via a highly polarized (and possibly ionized) TS, which is stabilized by water molecules via polar interactions or the formation of H-bonds (Mitroka et al., 2010). However, the electronegative moieties (here -OH) compete for electron density with OH, thereby lowering the stabilization of this polar TS by the water molecules. For the lower-MW AAs (and ethers) this effect is largely offset by the resonance stabilization of the TS (Mitroka et al., 2010), which likely becomes less important for the longer-chain molecules. These assumptions would explain the lower $k_{OH_{aq}}$ measured for the $C_5$-$C_{10}$ n-alcohols and α,ω-diols (curving of the plot in Fig. 5A), as compared with the corresponding n-alkanes.

The new kinetic data acquired in this work (Table 1) also indicates that the curving observed in the aqueous phase (Fig. 5A) is somewhat suppressed at higher temperatures for n-alcohols and especially for α,ω-diols (Fig. S6); naturally, diols can form a higher number of H-bonds with water than n-alcohols. Consequently, it is reasonable to assume that hydrophobic interactions (HI) can contribute to this phenomenon, including the formation of hydration spheres around the alkyl chains (Otto and Engberts, 2003). The H-atom transfer from organic reactants to OH is largely influenced by the number of water molecules (and the strength of H-bonds between them) in the first hydration shell (Yamabe and Yamazaki, 2018; Kroflič et al., 2020). At the same time, the stability of H-bonds between the water molecules in the hydration shell is decreased at higher temperatures (Ross and Rekharsky, 1996; Otto and Engberts, 2003; Chandler, 2005). For the longer chain AAs, with larger hydration shells, weakening of the H-bonds between the water molecules, and between water and -OH moieties (at T>298K) may render the aliphatic H-atoms more susceptible to abstraction by the OH. Naturally, in water, n-alkanes are also surrounded by H-bonded hydration layers, but the presence of -OH moieties can alter the hydration shells for the entire molecules (here AAs), which can explain the differences in the $k_{OH_{aq}}$ values for the $C_6$-$C_{10}$ n-alkanes, n-alcohols, and α,ω-diols (Otto and Engberts, 2003) – Fig. 5C.

Furthermore, when the $k_{OH_{aq}}$ and of $k_{OH_{gas}}$ values are compared directly, for the >$C_5$ molecules, n-alkanes undergo faster oxidation in water as compared with n-alcohols, whereas the opposite trend is observed in the gas phase for all $C_1$-$C_{10}$ homologs (Fig. 5C). In both gas and aqueous phases, OH most likely reacts with n-alkanes via the direct H-atom abstraction (Monod et al., 2005). In water, the resulting (highly polar) TS is stabilized by the solvent, which contributes to the higher $k_{OH}$ values for n-alkanes (Fig. 5C). However, for n-alcohols (Fig. 5C), the formation of the less energetically favorable TS (no H-bonded complex) in water most likely contributes to the observed decrease in the $k_{OH}$ values for the $C_4$-$C_{10}$ molecules. Also, for the lower-MW n-alcohols, the formation of the less energetically favorable TS in water may be partially offset by its resonance stabilization. Moreover, HIs can also contribute to the observed decrease in the $k_{OH}$ values for AAs in the aqueous phase, as compared with the gas phase (Fig. 5C).

The conclusions presented in this section indicate that, in the case of aliphatic, oxygenated molecules, the rates of R(I) in water may be largely influenced by the solvent effects: HIs and H-bonds formation. More often than not, such solvent effects will offset the "direct" activating effects of the oxygenated moieties, especially for the longer-chain molecules. However, further studies, likely involving computational methods, are needed to provide more quantitative conclusions.

The temperature-dependent values of $k_{OH_{aq}}$ measured in this work were used to derive activation parameters for the AAs under investigation (Table 2).

**Table 2.** The values of activation parameters obtained in this work

| Compound | A ×10$^{-11}$ (M$^{-1}$s$^{-1}$) | $E_A$ (kJ/mol) | $\Delta H^{\ddagger}$ (kJ/mol) | $\Delta S^{\ddagger}$ (J/mol×K) | $\Delta G^{\ddagger}$ (kJ/mol) | $R^{2,a}$ |
|---|---|---|---|---|---|---|
| Ethanol | 0.4±0.05 | 7.1±0.3 | 4.6±0.3 | -(51.2±1.1) | 19.9±0.5 | 0.984 |
| 1-Propanol | 0.8±0.1 | 8.5±0.4 | 6.0±0.4 | -(44.7±1.2) | 19.3±0.5 | 0.989 |
| 2-Butanol | 1.9±0.3 | 10.6±0.3 | 8.1±0.3 | -(37.5±1.1) | 19.3±0.5 | 0.994 |
| 1-Butanol | 2.5±0.3 | 10.7±0.3 | 8.3±0.3 | -(34.9±0.9) | 18.7±0.4 | 0.995 |
| 1-Pentanol | 6.2±1.3 | 12.1±0.5 | 9.7±0.5 | -(27.4±1.8) | 17.8±0.7 | 0.989 |
| 1-Hexanol | 8.2±2.6 | 12.6±0.8 | 10.1±0.8 | -(25.2±2.6) | 17.6±1.1 | 0.978 |
| 1-Heptanol | 14.4±6.1 | 13.9±1.1 | 11.4±1.1 | -(20.5±3.5) | 17.5±1.5 | 0.966 |
| 1-Octanol | 26.6±1.3 | 14.1±2.0 | 11.6±2 | -(19.4±6.8) | 17.4±2.8 | 0.944 |
| 1-Nonanol | 26.6±2.1 | 15.1±1.2 | 12.6±3.2 | -(15.4±7.2) | 17.2±1.6 | 0.968 |
| 1-Decanol | 32.8±2.6 | 15.6±2.6 | 13.1±2.6 | -(13.6±2.6) | 17.2±2.6 | 0.987 |
| 3-ethyl-3-pentanol | 6.4±2.1 | 13.8±0.8 | 11.3±0.8 | -(27.2±2.7) | 19.4±1.2 | 0.979 |
| 1,2-Ethanediol | 0.4±0.1 | 7.8±0.7 | 5.3±0.7 | -(49.6±2.4) | 20.1±1.0 | 0.951 |
| 1,2-Propanediol | 0.5±0.1 | 8.2±0.6 | 5.7±0.6 | -(48.6±2.2) | 20.2±0.9 | 0.964 |
| 1,2-Butanediol | 0.9±0.2 | 8.8±0.7 | 6.3±0.7 | -(43.9±2.3) | 19.4±1.0 | 0.966 |

| | | | | | | |
|---|---|---|---|---|---|---|
| 1,6-Hexanediol | 4.9±1.6 | 11.3±0.8 | 8.8±0.8 | -(29.5±2.8) | 17.6±1.2 | 0.968 |
| 1,7-Heptanediol | 7.8±2.8 | 12.2±0.9 | 9.7±0.9 | -(25.5±3.0) | 17.4±1.3 | 0.969 |
| 1,8-Octanediol | 31.8±4.5 | 15.6±1.1 | 13.2±1.1 | -(13.9±3.8) | 17.3±1.6 | 0.969 |
| 1,9-Nonanediol | 14.8±2.5 | 13.5±0.4 | 11.0±0.4 | -(20.2±1.4) | 17.0±0.6 | 0.994 |
| 1,10-Decanediol | 26.2±7.6 | 14.8±0.7 | 12.3±0.7 | -(15.5±2.4) | 16.9±1.0 | 0.986 |
| Cyclohexanol | 15.2±6.0 | 14.8±1.2 | 12.3±1.2 | -(20.0±3.8) | 18.3±1.6 | 0.965 |
| *trans*-1,2-Cyclohexanediol | 5.1±2.0 | 12.7±1.0 | 10.2±1.0 | -(29.1±3.2) | 18.9±1.4 | 0.966 |
| *exo*-Norborneol | 6.9±2.0 | 14.5±1.4 | 12.0±1.4 | -(26.6±4.8) | 19.9±2.0 | 0.943 |
| *cis*-2-Methylcyclohexanol | 18.4±5.1 | 14.5±1.4 | 12.1±1.4 | -(18.4±4.1) | 17.5±1.7 | 0.959 |
| (+)-Fenchol | 21.6±7.9 | 16.3±0.9 | 13.8±0.9 | -(17.1±3.0) | 18.9±1.3 | 0.982 |
| (+)-Borneol | 12.4±3.5 | 14.5±1.1 | 12.1±1.1 | -(21.7±3.6) | 18.5±1.5 | 0.967 |
| (−)-Menthol | 37.4±4.8 | 16.9±1.0 | 14.4±1.0 | -(12.5±3.3) | 18.1±1.4 | 0.980 |
| (±)-*exo,exo*-2,3-Camphanediol | 17.2±5.1 | 14.9±1.0 | 12.4±1.0 | -(19.0±3.4) | 18.1±1.4 | 0.972 |
| (1S,2S,3R,5S)-(+)-Pinanediol | 43.4±10.1 | 17.3±1.3 | 14.8±1.3 | -(11.3±4.2) | 18.2±1.8 | 0.966 |

[a]Squared coefficients of determination for the obtained Arrhenius plots

The values of $E_a$ obtained in this work for all AAs are much higher (Table 2) than the values for the same reactions in the gas phase, which also indicates a different, higher-energy TS in the aqueous phase (Mellouki et al., 2003; Monod et al., 2005). In the homolog series of n-alcohols and α,ω-diols, the values of $E_a$ increased (Fig. 6A and E), before reaching the average value of approx. 15 (kJ × mol$^{-1}$). Similar $E_a$ values were obtained for the nine cyclic AAs under investigation (Table 2). In addition to HIs (already discussed in this section), an increasing diffusion contribution (Table S3) can also contribute to the higher $E_a$ values for the longer-chain AAs (Schöne et al., 2014; Sarang et al., 2021). In solution, the rate of reaction is diffusion-limited if every encounter between reactants (here AAs and OH) leads to a reaction (Sarang et al., 2021). Consequently, the $E_a$ for the aqueous reactions with a high diffusion contribution is expected to approach 15 (kJ × mol$^{-1}$), likely due to the temperature-dependence of the viscosity of water, which also follows the Arrhenius relationship (Ervens et al., 2003). For all AAs under investigation, positive $\Delta H^{\ddagger}$ values, between 4 and 15 (kJ/mol) were obtained. In the homolog series of n-alcohols and α,ω-diols, the values of $\Delta H^{\ddagger}$ (Fig. 6B and F) follow the incremental increase of the $E_a$ values (eq. SI); a similar increase in the values of $\Delta S^{\ddagger}$ was observed (Fig 6C and G). For the majority of $C_1$-$C_{10}$ AAs, the literature (preferred) values of activation parameters also follow the general trends observed in this work, with the largest discrepancies observed for $C_2$-$C_4$ n-alcohols (Fig. 6A-C) and 1,2-ethanediol (Fig. 6G-H). For 1-butanol the recommended values of activation parameters (Figs. 6A-D) are based on a single study and the values derived in this work are closer to the data obtained by (Herrmann, 2003).

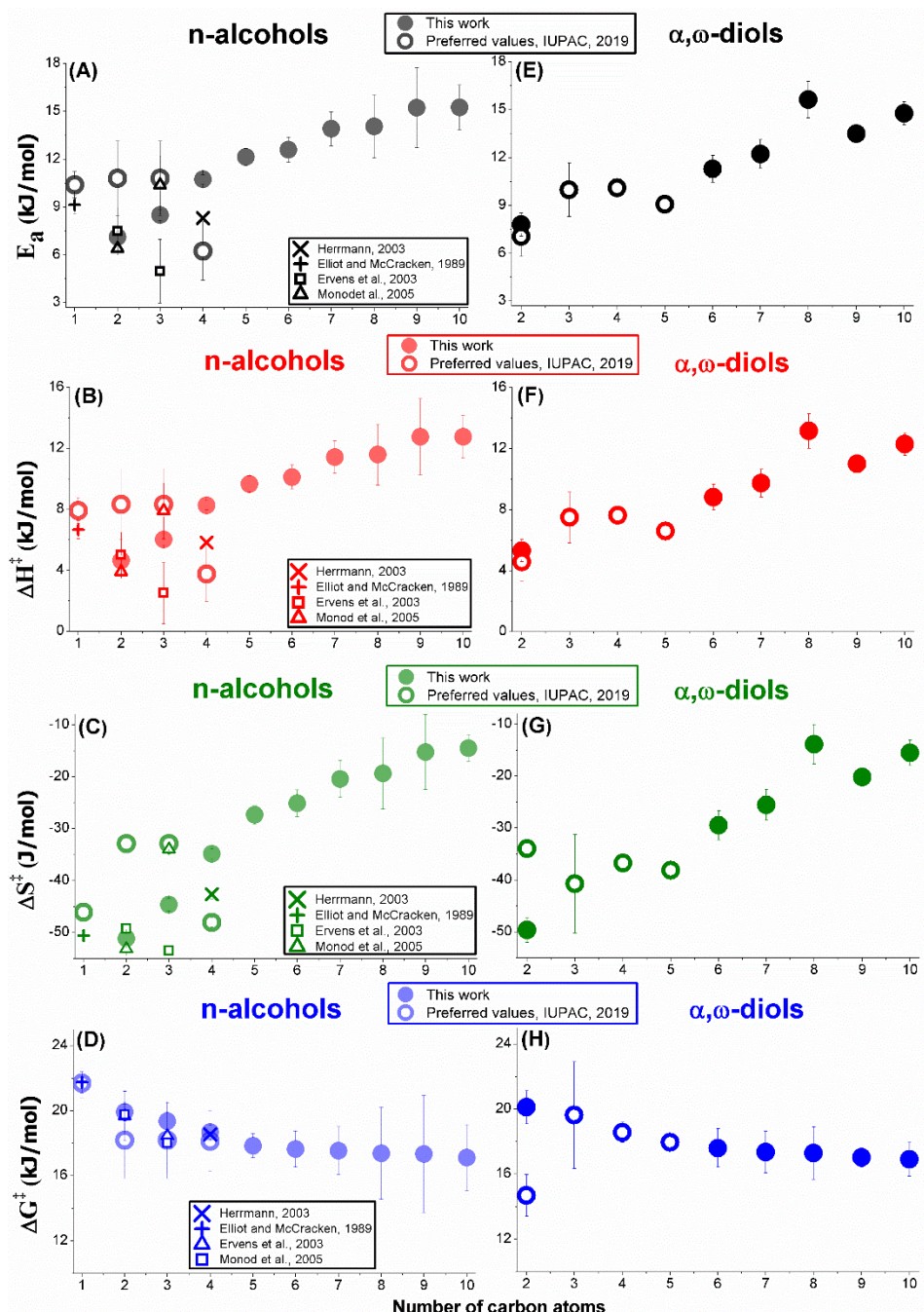

**Figure 6: The values of activation parameters (derived in this work and literature values) for the homolog series of C₁-C₁₀, linear n-alcohols, and α,ω-diols. The literature data shown are based on the preferred values (IUPAC, 2019); no recommendations currently are available for methanediol (formaldehyde hydrate).**

More experimental data are needed to provide a better estimate of the values of activation parameters for 1-butanol; analysis of the data available for 1,2-ethanediol (Fig.6G-H) leads to a similar conclusion (Hoffmann et al., 2009).

Negative values of $\Delta S^{\ddagger}$ were obtained for all AAs under investigation (Table 2). Higher (less negative) values of $\Delta S^{\ddagger}$ were obtained for the higher-MW AAs (including also cyclic AAs – Table 2), corresponding to a smaller decrease in disorder following R(I). Such a result may be associated with a release of water molecules from the hydrophobic surfaces (here alkyl chains) (Kroflič et al., 2020). As previously concluded, for some phenols, this phenomenon led to positive (or close to zero) values of $\Delta S^{\ddagger}$, even though the formation of a new bond between the reactants leads to a decrease in the disorder (Kroflič et al., 2020). The increase in the values of $\Delta S^{\ddagger}$ for the higher-MW AAs (with longer hydrophobic alkyl chains) also supports the assumption that the HIs influence the mechanism of R(I). The $\Delta G^{\ddagger}$ values (this work and literature data) are very similar (within the uncertainties reported - Fig. 6D and H) since the values of $\Delta H^{\ddagger}$ and $\Delta S^{\ddagger}$ underwent similar, incremental increases for all AAs under investigation (eq. SIII). Similar values of $\Delta G^{\ddagger}$ indicate the same TS and reaction mechanism (Ervens et al., 2003). The average $\Delta G^{\ddagger}$ of $18\pm2$ kJ/mol obtained in this work (Table 2) is close to the previously reported values, for the reaction of OH with the oxygenated, aliphatic molecules (Ervens et al., 2003; Hoffmann et al., 2009; Otto et al., 2018; Schaefer et al., 2020; Witkowski et al., 2021). Consequently, the values of activation parameters obtained in this work confirm a predominance of the H-atom abstraction mechanism (Ervens et al., 2003; Otto et al., 2018; Schaefer et al., 2020; Witkowski et al., 2021).

More patterns can be recognized by examining the $k_{OH}$ values for the linear, cyclic, and poly(alcohols) in both gas and aqueous phases. Some diols and poly(alcohols) with -OH moieties separated by more than one carbon atom exhibit noticeably higher values of $k_{OH_{aq}}$ as compared with their vicinal analogs; this trend can be observed for the $C_3$-$C_6$ AAs (Fig. 7A-D) (Hoffmann et al., 2009; IUPAC, 2019). The preferred $k_{OH_{aq}}$ values for 1-propanol (Fig. 7A), 1-butanol (Fig. 7B), and 1-pentanol (Fig. 7C) also follow this general trend. At the same time, results are somewhat inconclusive when the preferred $k_{OH}$ values for 1-hexanol and 1,6-hexanediol are considered, due to their relatively high uncertainties (Fig. 7D). In the gas phase, the $k_{OH_{gas}}$ for the vicinal butanediols are lower as compared with the 1,4-butanediol, but this conclusion is again somewhat implied by the statistical data (Fig. 7E). Generally, an insufficient number of the $k_{OH_{gas}}$ are available for diols to present more definitive conclusions (McGillen et al., 2021).

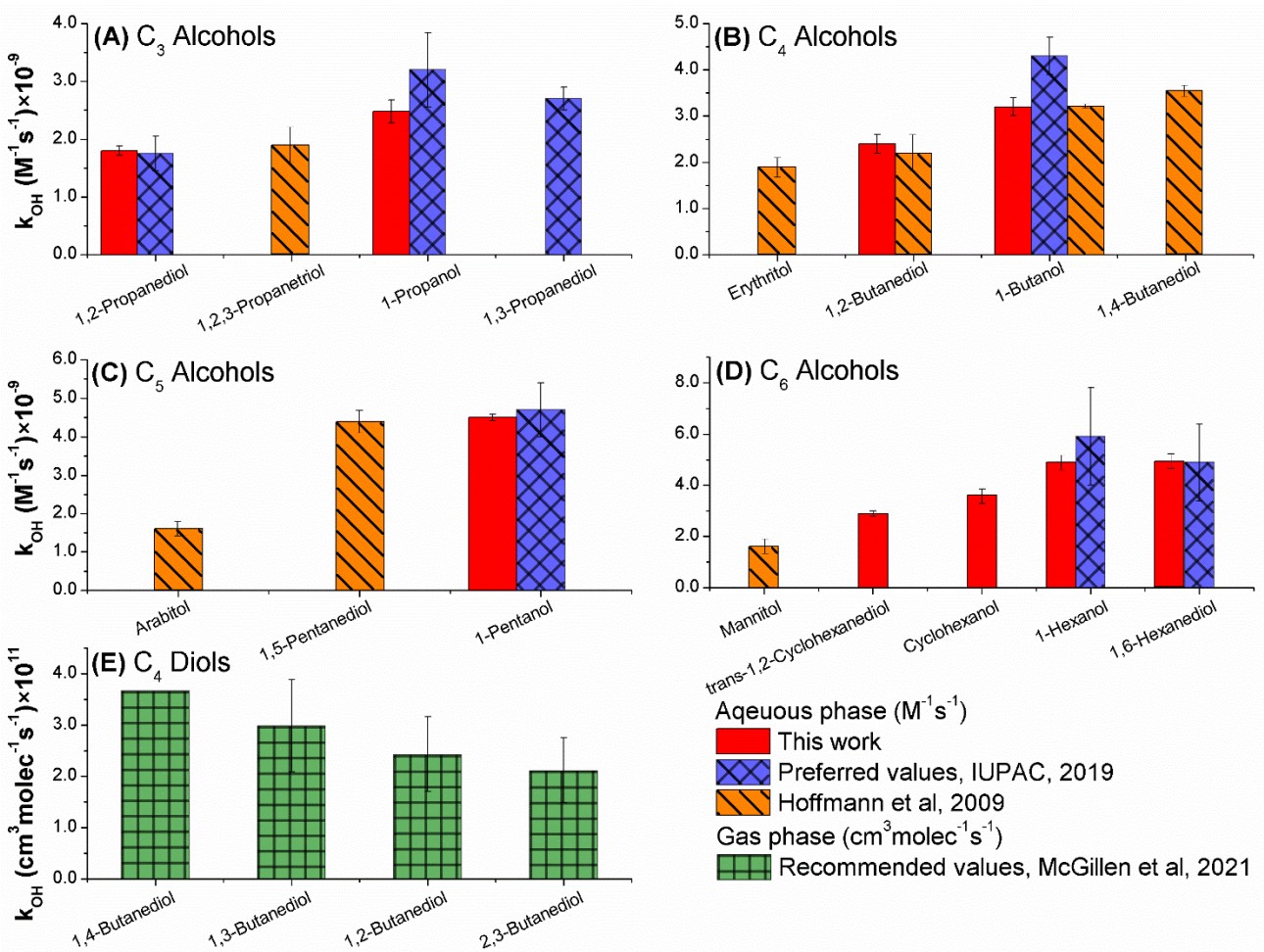

**Figure 7: Measured $k_{OH}$ values (this work and literature data) for the selected mono and poly(alcohols). When available, the preferred (recommended) values are shown (IUPAC, 2019; M. R. McGillen, 2021). For the three sugar alcohols (erythritol, arabitol, mannitol), the data reported by (Hoffmann et al., 2009) are presented.**

Under the assumption that the trends illustrated in Fig. 7, are not related directly to the solvent effects (hence are also observed in the gas phase - Fig. 7E), the lower $k_{OH}$ values for the vicinal poly(alcohols) and diols may be rationalized by considering their structural features. In 1,2-diols the -OH moieties most likely interact via polarization and electrostatic effects, rather than forming intramolecular hydrogen bonds (Klein, 2003; Das et al., 2015). Such interactions may affect the mesomeric donor effect of -OH (Monod and Doussin, 2008), thereby resulting in the observed decrease in the measured $k_{OH_{aq}}$ values for the vicinal poly(alcohol)s.

The experimental data acquired in this work, and the literature data (Buxton et al., 1988; McGillen et al., 2020; M. R. McGillen, 2021), also revealed noticeably lower $k_{OH_{aq}}$ values for some cyclic AAs and hydrocarbons as compared with their linear analogs (Fig. 8A-C). This trend is especially noticeable for exo-norborneol, which possesses a bridged cyclohexane ring,

(effectively two fused $C_5$ rings). Note however that the preferred $k_{OHaq}$ values for $C_6$ and $C_7$ linear AAs and diols, are the same as compared with their cyclic analogs, within the reported uncertainties (IUPAC, 2019).

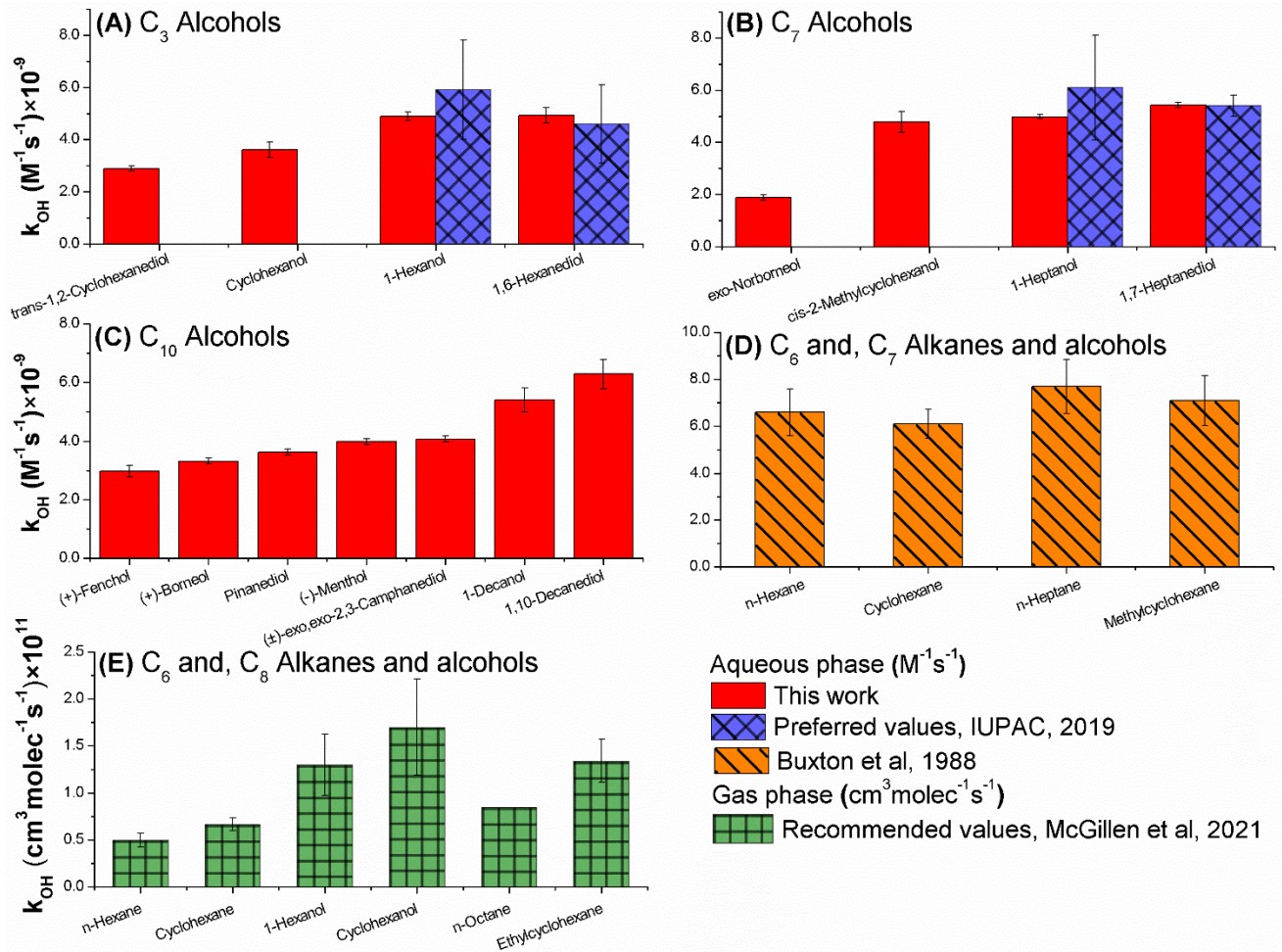

**Figure 8: $k_{OH}$ values at 298K for the cyclic and linear alcohols and diols in the aqueous (A-C) and gas (D, E) phases. When available, the preferred (recommended) values are shown (IUPAC, 2019; M. R. McGillen, 2021). For the sugar alcohols, the data reported by (Hoffmann et al., 2009) presented. For 1-hexanol and 1,6-hexanediol (A, B) both the preferred values and (IUPAC, 2019), the data acquired in this work is included.**

For alkanes, the $k_{OHaq}$ are the same (within the uncertainties reported) for both cyclic and linear molecules (Fig. 8D). Furthermore, the $k_{OHgas}$ values are the same for the n-hexane and cyclohexane (and also for the corresponding alcohols), within the uncertainties reported (Fig. 8E). Also, no uncertainty has been reported for the $k_{OHgas}$ value for n-octane; for this reason, it is not possible to evaluate whether or not the $k_{OHgas}$ value for n-octane differs significantly from ethylcyclohexane (Fig. 8E).

The lower $k_{OH_{aq}}$ values for some cyclic AAs may be caused by the different mobilities of the linear and cyclic reactants inside the water cage, resulting in a different intramolecular selectivity of OH (Kopinke and Georgi, 2017). However, more experimental data are needed to provide more insights into the reactivities of the aliphatic, oxygenated molecules containing aliphatic rings; for instance, the $k_{OH_{aq}}$ values measured in this work for *cis*-2-methylcyclohexanol, 1-heptanol and 1,7-heptane diol are the same; the preferred $k_{OH_{aq}}$ values for the $C_7$ AAs also support this conclusion (Fig. 8B).

### 3.2 Structure-activity relationship

The $k_{OH_{aq}}$ values measured in this work at 298K, and the literature data (section 2.5), were used to optimize F and G factors only for $CH_2$ and -OH moieties (Model 1) or substituent factors for $CH_3$, $CH_2$, CH, and -OH, moieties, $C_4$-$C_6$ rings and the base (partial) rate coefficients (Model 2) – Table 3. Model 2 also included a new substituent factor for molecules with straight-chain carbon backbones - F $\geq(CH_2)_6$ - Fig. 1.

**Table 3.** SAR (Model 2) substituent factors and base rate coefficients

| Group | Substituent factors | | Ref. |
|---|---|---|---|
| | F | G | |
| $CH_3$ | 1.40 | 1.16 | This work |
| $CH_2$ | 1.34 | 1.00 | This work |
| CH | 1.03 | 1.05 | This work |
| $\geq(CH_2)_6$ | 0.76 | | This work |
| C | 1.00 | 1.00 | (Monod and Doussin, 2008) |
| OH | 1.97 | 0.56 | This work |
| $C_6$ | 0.65 | | This work |
| $C_5$ | 0.61 | | This work |
| $C_4$ | 1.34 | | This work |
| FC=O | 0.20 | 0.90 | (Doussin and Monod, 2013) |
| COOH | 0.16 | 0.59 | (Witkowski et al., 2021) |
| COO- | 0.54 | 0.64 | (Witkowski et al., 2021) |
| **Base (partial) rate coefficients, ($k_{OH}$) at 298K×$10^{-8}$** | | | |
| $k_{CH3}$ | 4.06 | | |
| $k_{CH2}$ | 6.17 | | This work |
| $k_{CH}$ | 4.40 | | |
| $k_{OH}$ | 1.54 | | |

| Temperature-dependence of the base rate coefficients | | | |
|---|---|---|---|
| | $C\times10^{-3}$, $(M^{-1}s^{-1})$ | $D\times10^{-2}$, (K) | |
| $k_{CH3}$ | 5.50 | 0.55 | |
| $k_{CH2}$ | 197.9 | 9.98 | This work |
| $k_{CH}$ | 5.04 | 0.05 | |
| $k_{OH}$ | 1.74 | 0.01 | |

The unadjusted SAR included the neighboring parameters and partial rate coefficients originally derived by (Monod and Doussin, 2008; Doussin and Monod, 2013), and F and G factors for carboxylic acids and carboxylate ions recently updated by
405 our group (Witkowski et al., 2021). Note also that *cis*-pinonic acid was the only carbonyl compound in the kinetic dataset used in this work to optimize SAR parameters (Witkowski, 2023). As demonstrated in our previous study, hydration (formation of gem-diol) of *cis*-pinonic acid is negligible (Witkowski et al., 2021).

In Table 3, the F and G values >1 indicate activation whereas values <1 correspond to the deactivating effect of a given group (Kwok and Atkinson, 1995). The values of F and G factors are often rationalized via the field and resonance effects, which are
410 expressed by the values of (*R,* resonance) and (*F,* field) parameters defined by (Swain and Lupton, 1968). The resonance effects primarily affect the α-position whereas the field effects are mostly associated with the β-positions (Monod and Doussin, 2008). In Table 4, the negative values correspond to the electron-donating group (activating effect) whereas positive values indicate deactivation (electron withdrawing). Consequently, alkyl substituents should exhibit activating effects in both α and β-position following their negative values of F and R parameters, whereas α-activation and β-deactivation are expected for -
415 OH (Table 4).

**Table 4.** Selected resonance (R) and field (F) factors; negative values correspond to activating effects and positive values indicate deactivation

| Group | Resonance effects (R, α-position) | Field effects (F, β-position) |
|---|---|---|
| -OH | -1.89 | 0.46 |
| CH$_3$ | -0.41 | -0.1 |
| CH$_3$CH$_2$ | -0.44 | -0.2 |

In Model 1, the F and G factors of 1.40 (F), 0.63 (G) for -CH$_2$, and 1.51 (F) and 0.81(G) for -OH were obtained, what only
420 resulted in a moderate improvement of the SAR accuracy (Fig. 9B), even though a majority of model compounds (Table 1) possessed both of these moieties, which should result in a reliable model fit (Peduzzi et al., 1996).

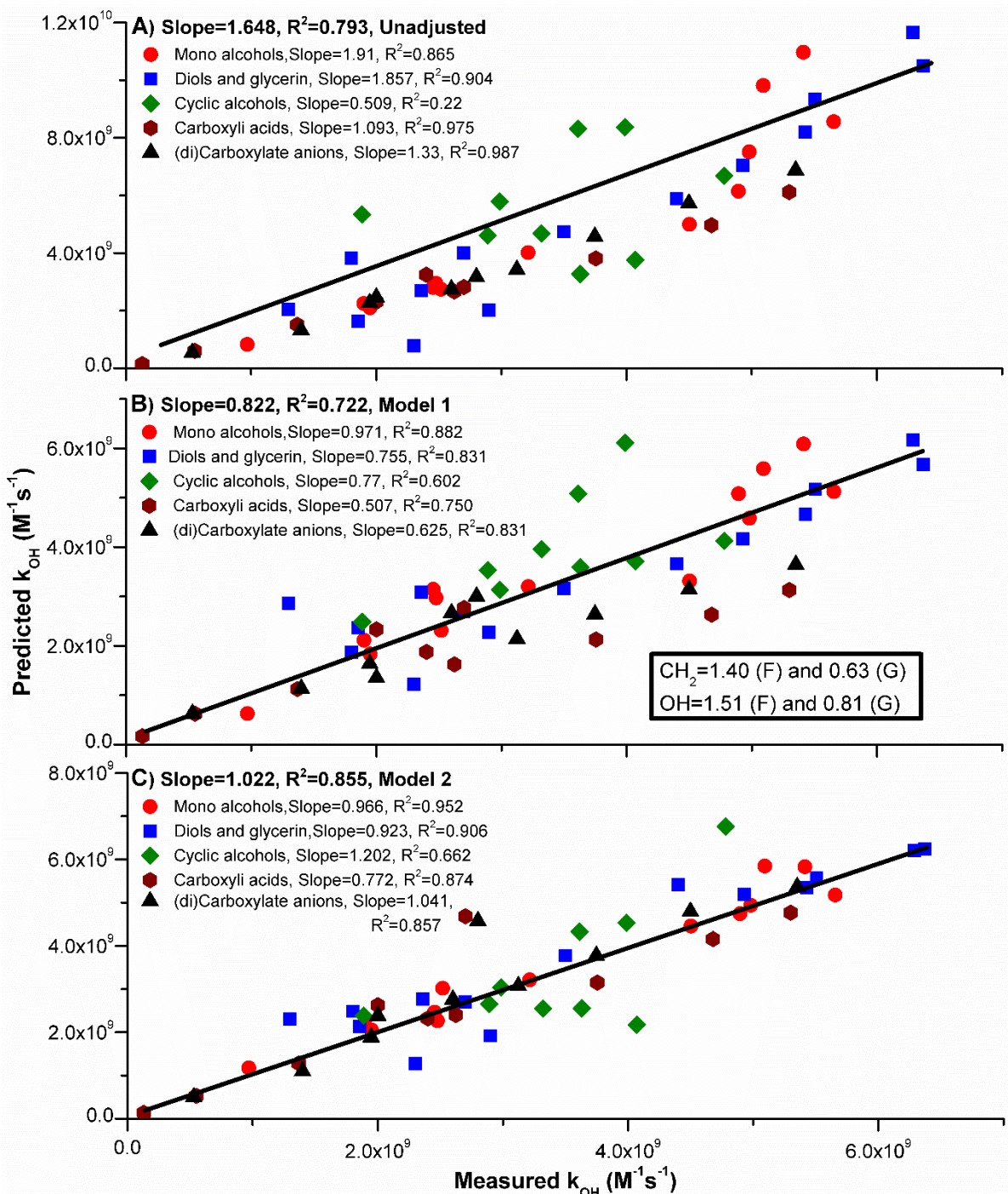

**Figure 9: Accuracies of unadjusted (A), and adjusted (B, Model 1) and (C, Model 2) SARs for alcohols and carboxylic acids at 298K.**

The values obtained for -OH (Model 1) are similar to the previously reported values (Monod and Doussin, 2008). but the strong β-deactivating effect of $CH_2$ (Fig. 9B) is difficult to rationalize in the context of the field and resonance effects for alkyl

substituents (Table 4). Such behavior of Model 1 most likely reflects the trends in OH reactivities observed for the homolog series of linear alcohols and diols in the aqueous phase (Figs. 4 and 5).

By only adjusting the F and G factors for -OH and CH$_2$, it was impossible to further improve the performance of SAR. Moreover, the accuracy of Model 1 was higher for AAs but significantly lower for carboxylic acids and carboxylate ions (Fig. 9B). In an attempt to further improve the model, all substituent factors and partial rate coefficients were adjusted. The resulting SAR parameters showed very little correlation with the factors proposed by Swain and Lupton (results not shown). At the same time, only a minor improvement in the SAR performance was achieved, but the obtained G-parameter values for the CH$_2$ group were always <1.

The original SAR developed by Monod and Doussin was based on the values of $k_{OH_{aq}}$ measured primarily for the lower-MW molecules, which is reflected by the large positive bias of the unadjusted model (slope=1.648, Fig. 9A), especially for the longer-chain AAs (Monod and Doussin, 2008; Doussin and Monod, 2013). Consequently, using this methodology, which was originally developed for predicting the values of $k_{OH_{gas}}$, an adequate representation of the aqueous OH reactivity of higher-MW molecules in the aqueous phase is difficult.

For this reason, it was concluded that there is a need for an additional parametrization of the observed reactivity of the WSOCs with larger carbon backbones (Fig. 5). Therefore, a new F factor, reflecting the reactivities of the longer chain molecules, was introduced (Table 3). In Model 2, the partial k$_{OH}$ of every CH$_2$ moiety is additionally adjusted for the C$_6$ and longer-chain precursors, which resulted in a noticeably higher accuracy of SAR (Fig.9C). Following this adjustment, the versatility, and accuracy of Model 2 were noticeably improved, resulting in the better performance for alcohols, diols and carboxylic acids and carboxylate anions (Fig. 9C). Accuracy of SAR (Model 2) was lowest for cyclic alcohols (Fig. 9B). The F-factors <1 obtained for the C$_5$ and C$_6$-rings reflect the lower OH reactivates observed in this work for the cyclic AAs (Fig. 8).

In Model 2, the adjusted values of the neighboring parameters and partial rate coefficients for the alkyl groups (CH$_3$, CH$_{2,}$ and CH) – Table 3 - are all in good agreement with the previously reported data (Monod and Doussin, 2008). Furthermore, the substituent factors derived for -OH are also in very good agreement with the literature data and correlate well with the resonance and field effects of the hydroxyl moiety (Table 4). The value of the partial rate coefficient for the abstraction of H-atom from the -OH was approx. three times lower than the rates obtained for the CH$_n$ groups (Table 2), thereby confirming that a direct abstraction of H-atom from -OH is a minor reaction pathway (Atkinson, 1986b; Kwok and Atkinson, 1995; Monod and Doussin, 2008).

The optimized SAR (Model 2) was used to obtain the values of C and D factors (eq. IV and V), which allows for predicting the temperature-dependent $k_{OH_{aq}}$ values for AAs and carboxylic acids induced in the model training set (Fig. 10).

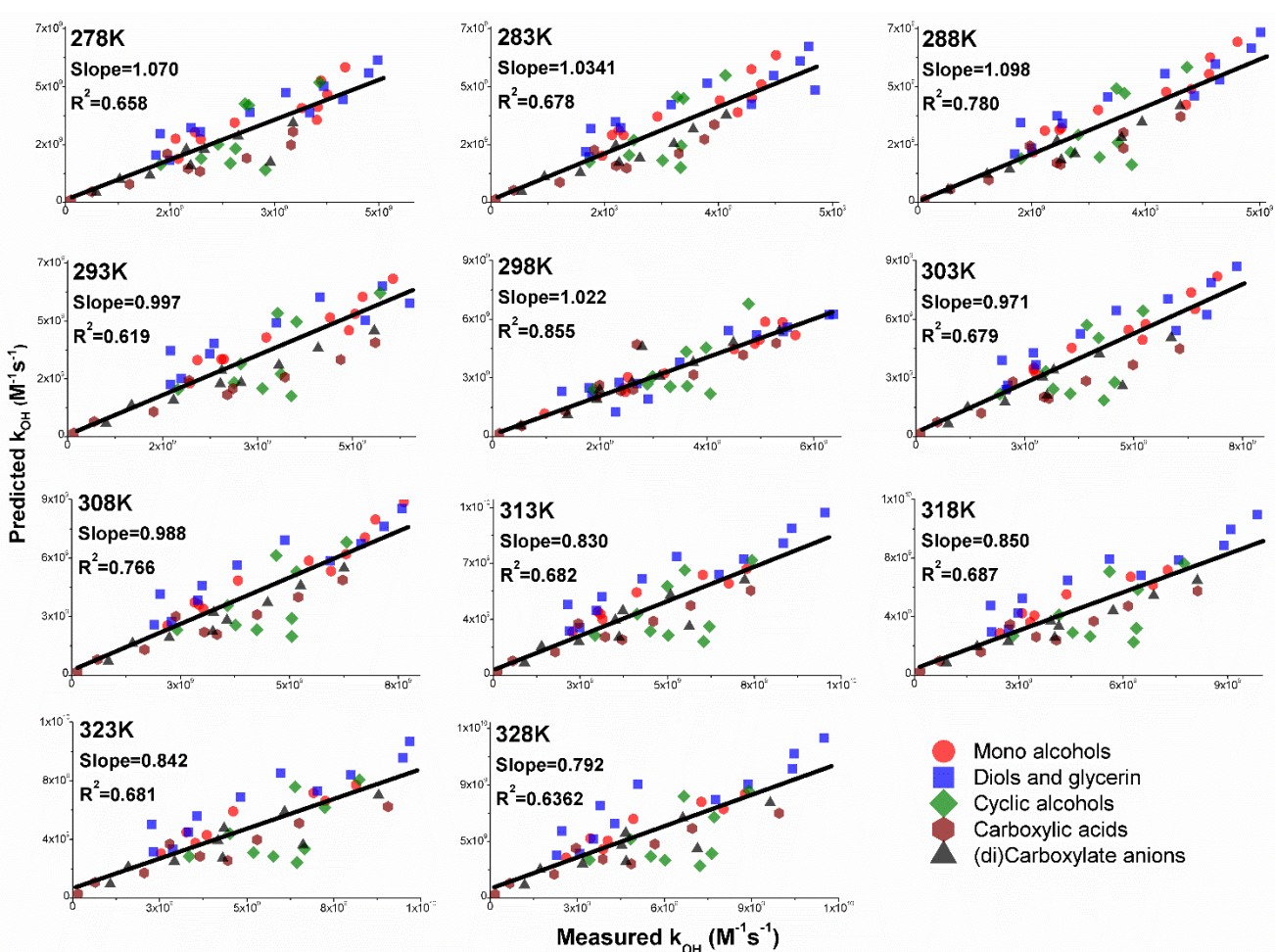

**Figure 10: Accuracies of Model 2 for alcohols and carboxylic acids in the temperature range between 278 and 328K. If for a given temperature, the measured value of $k_{OH_{aq}}$ was not available at a given temperature, it was derived using the reported (or measured in this work) values of $E_a$ and A – see also Appendix 1.**

The overall performance of Model 2 was similar, for alcohols and carboxylic acids under investigation, in the temperature range between 278-328 K; the best performance was obtained for 298K, likely because the values of neighboring parameters were optimized using the values of $k_{OH_{aq}}$ measured at this temperature (section 2.5 and eq. III). The values of squared coefficients of determination ($R^2$) and slopes obtained from the linear regression analysis for different groups of compounds (Fig. 10) are presented in Fig. 11 – these data are listed in Table S9.

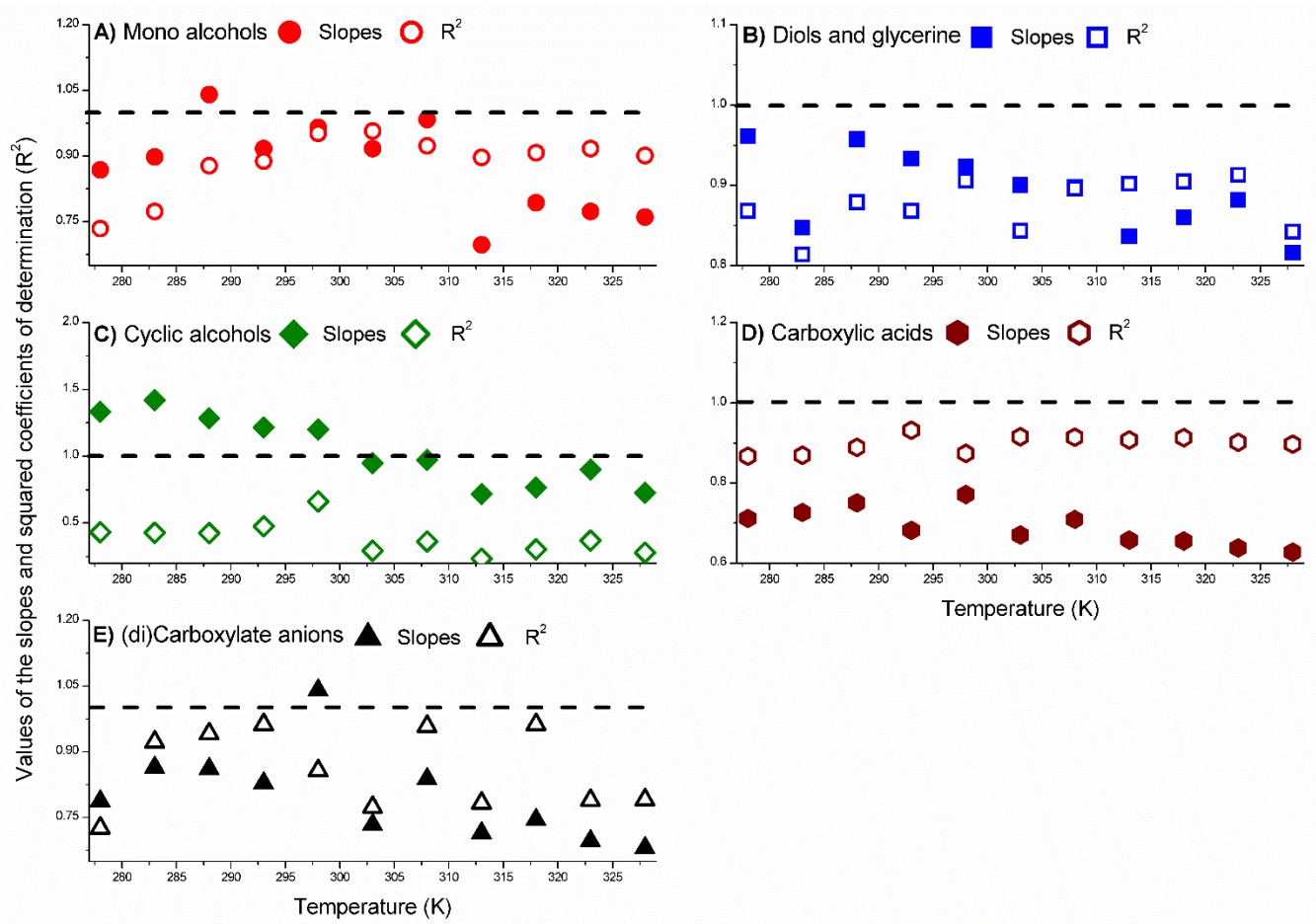

**Figure 11: The values of slopes and squared coefficients of determination ($R^2$) obtained from the linear regression analysis of the performance of Model 2 for mono alcohols (A), diols and glycerine (B), cyclic alcohols (C), carboxylic acids (D) and carboxylate anions (E) in the temperature range between 278-328K. The dotted horizontal lines represent an ideal model; slope and $R^2$=1.**

The accuracy of Model 2 in the studied temperature range was satisfactory for AAs, diols, carboxylic acids, and carboxyla anions. The model showed a slight negative bias at temperatures >308 K for mono alcohols (Fig.11A), diols, and glycerine

(Fig.11B). Furthermore, the correlation between the measured and predicted $k_{OH_{aq}}$ values was lower for both higher and lower temperatures. This may be due to the non-linear effects of the temperature on the $k_{OH_{aq}}$ values observed for the n-alcohols and α,ω-diols (section 3.1). Similar performance of Model 2 was also obtained for carboxylic acids (Fig.11D) and carboxylate anions (Fig.11E) and, the model performance was unsatisfactory for the cyclic AAs investigated in this work (Fig.11C). However, it is important to note that in the entire temperature range investigated in this work, the vast majority of $k_{OH_{aq}}$ values

for cyclic alcohols were predicted within a factor of 2.

## 4 Atmospheric implications

### 4.1 Lifetimes of terpenoic alcohols

The lifetimes for the five TAs investigated in this work were estimated with eq. VII (Sarang et al., 2021).

$$\tau = \frac{1}{\left(\dfrac{k_{OH_{gas}}}{H_{OH}^{cc}} + k_{OH_{aq}} H_{AA}^{cc} \omega\right)[OH]_{aq}} \qquad (VII)$$

In eq. VII, $\tau$ is the combined lifetime in the gas and the aqueous phase due to the reaction with the OH. Therefore, this approach considers both gas and aqueous oxidation of AAs by the OH and their relative importance, depending on Henry's law equilibria and LWC of a given air mass (Sarang et al., 2021).

In eq. VII, $k_{OH_{gas,}}$ and $k_{OH_{aq}}$ are the bimolecular reaction rate coefficients for the reaction of a given AA in the gas and aqueous phase with the OH (M$^{-1}$s$^{-1}$) at 298K respectively. $H_{OH}^{cc}$ and $H_{AA}^{cc}$ are the dimensionless Henry's law constants for the OH ($H_{OH}^{cc}$) and for the given AA ($H_{AA}^{cc}$) (Sander, 2015), $\omega$ is the liquid water content (LWC, m$^3$/m$^3$), $[OH]_{aq}$ is the aqueous concentration of OH (M). The gas-phase OH concentration is connected with $[OH]_{aq}$ via Henry's equilibrium, assuming that there are no additional sources of OH; the average reported $H_{OH}^{cc} = 764$ was used (Sander, 2015).

**Table 5.** Data used to estimate the atmospheric lifetimes of the five terpenoic alcohols due to reaction with the OH at 298K

| Name | $k_{OH_{aq}} \times 10^{-9}$ $(M^{-1}s^{-1})^a$ | $k_{OH_g} \times 10^{11}$ $(cm^3 molec^{-1}s^{-1})$ | Ref. | $H^{cc}$ | Ref. |
|---|---|---|---|---|---|
| Fenchol | 2.99 | 2.49 | (McGillen et al., 2020; McGillen et al., 2021) | $5.0 \times 10^3$ | (Fichan et al., 1999) |
| Borneol | 3.32 | 2.65 | | $5.08 \times 10^3$ | (Sander, 2015) |
| Menthol | 3.99 | 1.48 | | $1.20 \times 10^3$ | (Sander, 2015) |
| Camphanediol | 4.07 | 2.78 | (Atkinson, 1986a; Kwok and Atkinson, 1995) | $1.0 \times 10^5$ | |
| Pinanediol | 3.63 | 2.12 | | | (EPISuite4.11) |
| | | | | $1.0 \times 10^5$ | |

$^a k_{OH_{aq}}$ values measured in his work

The $k_{OH_{aq}}$ values measured in this work were used to calculate the $\tau$ for the five TAs. For pinanediol, the $k_{OH_{gas}}$ value was predicted with SAR due to a lack of measured values. Likewise, when possible, the experimentally measured $k_{OH_{gas}}$ and $H_{AA}^{cc}$ values were used; otherwise, the values listed were estimated with structure-activity relationship (SAR) (Atkinson, 1986a; Kwok and Atkinson, 1995), and with HenryWin - Table 5.

The $\tau$ values were calculated as a function of LWC and $[OH]_{aq}$; the dotted lines in Fig. 12 represent the range of OH concentrations in different types of aqueous particles (Herrmann et al., 2010). The y-axis (Fig. 12) represents the total lifetime

$(\tau, h)$. The plateau regions, corresponding to the lower values of LWC, indicate that the reaction with the OH takes place only in the gas phase.

When this simplified model (eq. VII) is used, the time scales of interactions of gaseous species (air masses) also have to be considered. An average, weighted residence time of an air parcel inside a cloud was estimated as approx. 3h; by taking into
505 account the total lifetime of a typical air parcel (~ 5 days), it will spend a total of 18 h inside a cloud (Herrmann et al., 2015). Furthermore, the aqueous-phase processing of organic molecules (here TAs) in the atmosphere is interrupted by the evaporation of cloud and fog droplets, which occurs on a time scale of a few minutes (Ervens, 2015).

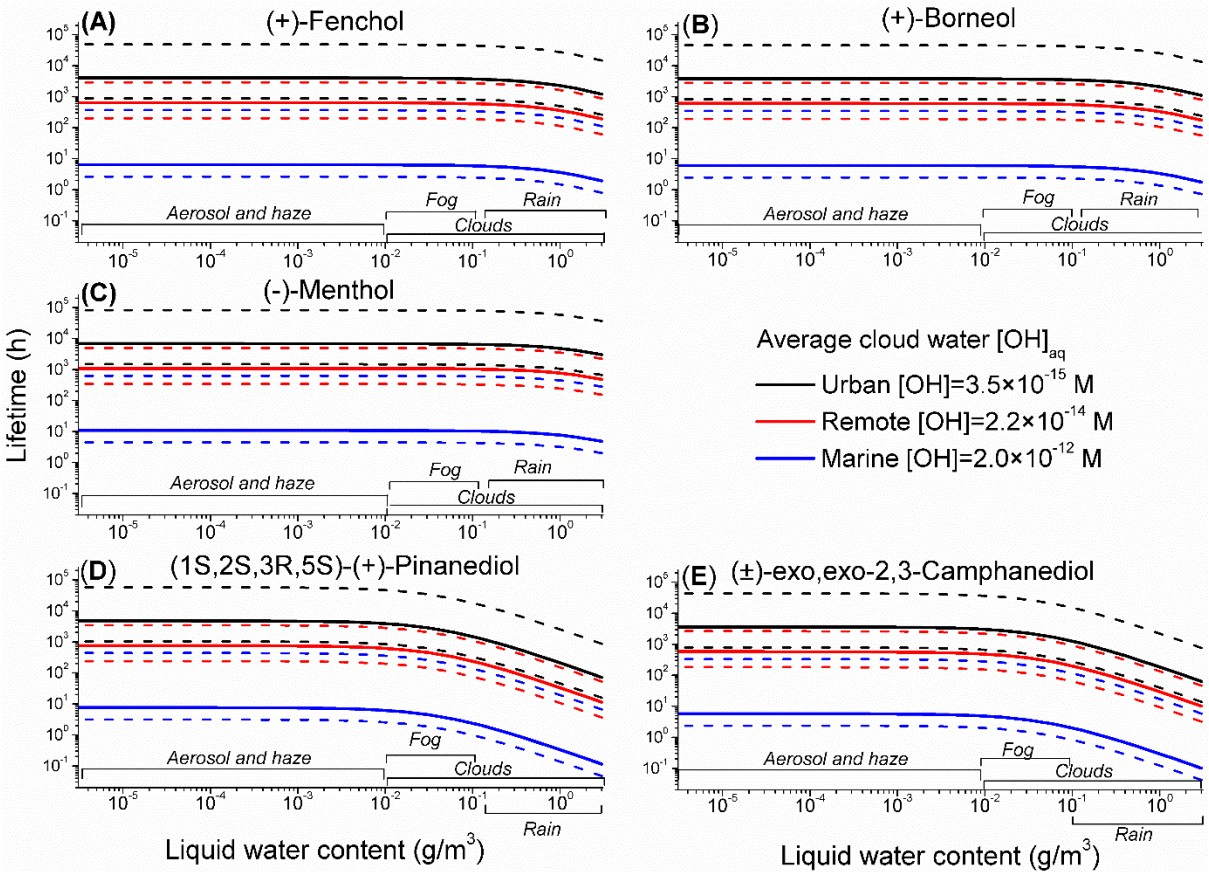

**Figure 12: The estimated lifetimes of TAs. The dotted lines represent lifetimes corresponding to the minimum and maximum [OH]$_{aq}$**
**in different aqueous particles.**

The new $k_{OH_{aq}}$ values acquired in this work indicate that TAs are likely to undergo aqueous oxidation in clouds with LWC$\geq$0.1 g/m$^3$ (Fig. 12A-C) whereas terpenoic diols will react with the OH in aqueous aerosols, clouds, and fogs with LWC$\geq$10$^{-4}$ g/m$^3$ (Fig.9 D-E). Considering the time scale of air-parcel interactions with clouds and fogs and, the lifetimes of aqueous particles, all TAs under investigation can undergo R(I) in maritime clouds (Fig. 12). At the same time, aqueous oxidation of TAs over
515 the ocean is only plausible at the ocean side (Hiura et al., 2021) whereas in marine aerosols, the dominant AAs are fatty and

sugar alcohols (Fu et al., 2013). Only terpenoic diols can undergo R(I) in both continental (here urban and remote) as well as in marine clouds, with estimated lifetimes <1min; well below the estimated lifetimes of cloud and fog droplets (Fig. 12D and E).

The aqueous oxidation of terpenoic acids by OH has been receiving considerable attention in connection with the formation of $_{aq}$SOAs (Aljawhary et al., 2016; Zhao et al., 2017; cis-Pinonic Acid Oxidation by Hydroxyl Radicals in the Aqueous Phase under Acidic and Basic Witkowski and Gierczak, 2017; Otto et al., 2018; Witkowski et al., 2018b; Witkowski et al., 2018a, 2019; Amorim et al., 2020; Amorim et al., 2021; Witkowski et al., 2021). Using the same approach (eq. VII) we have recently derived τ values for six terpenoic acids, identified as the major components of unaged α-pinene SOA (Witkowski et al., 2023). It was concluded that *cis*-pinic, *cis*-pinonic, hydroxypinonic, and oxopinonic acids will undergo a reaction with the OH in the aqueous phase LWC is ≥ 1×10$^{-3}$ (g/m$^3$). Hence, in the case of terpenoic diols (Fig. 12), their potential to undergo aqueous oxidation under realistic atmospheric conditions is comparable with some terpenoic acids. In the case of *cis*-pinonic acid, the measured mass yields of $_{aq}$SOAs following the reaction with the OH were as high as 60%. (Aljawhary et al., 2016) At the same time, to date, little mechanistic data is available about the formation of low-volatility products from TAs following R(I).

Global emissions of oxygenated terpenes are tentatively estimated at 26 (TgC×yr$^{-1}$) (Sindelarova et al., 2014). TAs are emitted by vegetation (Héral et al., 2021), but are also used in the industry, as fragrance agents and solvents.(Belsito et al., 2008) Therefore, TAs, and especially terpenoic diols, are potentially important precursors of $_{aq}$SOAs in both urban and remote environments.

## 4.2 Influence of the temperature

The values of $H^{cc}$, $k_{OH_{aq,}}$ and $k_{OH_{gas}}$ all depend on the temperature. In a relatively narrow temperature range, in which liquid water exists in the atmosphere, the enthalpy of dissolution $\left(\frac{dln(H)}{d1/T}\right)$ can be considered constant. Hence, the Van't Hoff equation can be applied to describe the temperature dependence of $H^{cc}$ values (Sander, 2015). For instance, a decrease in the temperature by 20° will result in a 2 to 7-fold increase in the $H^{cc}$ values of TAs and OH (Sander, 2015). Consequently, a significantly higher fraction of these reactants will reside in the aqueous phase at lower temperatures, thereby strongly favoring the aqueous chemical aging of TAs in the atmosphere.

The data acquired in this work, and the literature data confirm that the temperature dependence of the $k_{OH_{aq}}$ values for the TAs follow the Arrhenius relationship (Witkowski, 2023). Considering the temperature range in which liquid water exists in the atmosphere, the same conclusion also applies to the temperature dependence of the $k_{OH_{gas}}$ values for TAs and similar molecules. Furthermore, the values of E$_a$ and A (section 2.4) measured in this work, as well as literature data, show relatively similar temperature dependencies of $k_{OH_{aq}}$ and $k_{OH_{gas}}$ (Atkinson, 1986b; Kwok and Atkinson, 1995; Herrmann, 2003; Hoffmann et al., 2009; McGillen et al., 2020). Hence, the $k_{OH_{aq}}/k_{OH_{gas}}$ ratio for the TAs (Table 5) will remain mostly

unchanged in the considered temperature range. At the same time, a lower temperature will result in a lower oxidation rate, thereby decreasing the τ values, which can make other removal mechanisms more relevant.

These conclusions are valid primarily for cloud water droplets which are diluted and, more often than not, can be treated as ideal solutions (Herrmann, 2003; Richters et al., 2015). On the other hand, the ionic strength (I) of some aerosols can be as high as 43 (M) (Cheng et al., 2016). The high I values of some aerosols will affect the kinetics of R(I), partitioning, and rates of diffusion of the reactants (Mekic and Gligorovski, 2021). While salting-out of organics will likely occur in aqueous particles with high I value, salts are also expected to enhance the rate of R(I) (Zhou et al., 2019).

## 4.4 Potential formation of $_{aq}$SOAs from terpenoic alcohols

No product studies were carried out in this work. Nevertheless, as indicated by the new kinetic data acquired in this work, the cyclic TAs investigated should be considered as the potential precursors of $_{aq}$SOAs. For instance, cyclic terpenes are often more efficient at forming SOAs as compared to linear precursors (Lim and Ziemann, 2009; Hunter et al., 2014). Following RI, oxygenated functional groups are added to the original carbon backbone of the precursor, according to Russell's and Bennett-Summer's mechanisms (Russell, 1957; Bennett and Summers, 1974). Alkoxy (RO) radicals are formed from the reaction of an alkyl radical with $O_2$ via the decomposition of tetroxide intermediates (Bennett and Summers, 1974; von Sonntag and Schuchmann, 1991). At the same time, in the case of linear precursors, the β-scission reaction of RO radicals can result in a decreased length of the carbon backbone (Rauk et al., 2003; Enami and Sakamoto, 2016; Murakami and Ishida, 2017). Consequently, β-scission involving non-cyclic precursors results in lower MW and, more volatile products, that are less likely to contribute to SOAs (Rauk et al., 2003; Enami and Sakamoto, 2016; Murakami and Ishida, 2017). Hence, the cyclic precursors, such as TAs, are more likely to yield highly-oxidized, low-volatility following reaction with the OH without undergoing too much fragmentation (Claeys et al., 2009; Ceacero-Vega et al., 2012; Sato et al., 2016).

## 5 Conclusions and further work

The use of a bulk photoreactor combined with the off-line, semi-quantitative analysis of the reacting WSCOs via gas chromatography allowed us to obtain $k_{OH_{aq}}$ for multiple compounds from a single experiment (Herrmann et al., 2005; Monod et al., 2005). This method is significantly less laborious as compared with some of the other methods of measuring $k_{OH_{aq}}$, and can be used to obtain large kinetic datasets, thereby facilitating the expansion of the aqueous kinetic databases for the environmentally widespread WSOCs.

The measured $k_{OH_{aq}}$ values for AAs (this work and literature data) revealed that it was necessary to introduce an additional neighboring parameter for the linear, longer chain ($\geq C_6$) molecules to further improve the performance of the predictive model. The need to modify the methodology used in the original SAR was most likely connected with a different TS formed between OH and AAs in the gas and the aqueous phases (Smith and Ravishankara, 2002; Mellouki et al., 2003; Herrmann et al., 2005).

It is still unclear if the new, modified SAR is capable of accurately predicting the values of $k_{OH_{aq}}$ for different oxygenated molecules, including carbonyls, esters, linear acids, and multifunctional molecules, possessing larger carbon backbones. More kinetic data for such molecules, including polyfunctional compounds, are needed to further improve the capabilities of kinetic predictive models (SARs).

Models utilized for generating explicit oxidation schemes of atmospherically abundant organics frequently rely on kinetic SARs to derive rate coefficients and branching ratios of the numerous by-products involved in a given mechanism (Bräuer et al., 2019). Hence, expanding the applicability domains of the aqueous kinetic SARs will further improve our ability to predict the effects of various, multiphase processes (like for instance formation and aging of aqSOAs) on air quality, climate, and public health. Likewise, the dependence of the $k_{OH_{aq}}$ values from I also needs to be further investigated to provide insights

into the time scales of R(I), and other radical reactions, in the presence of high amounts of salts (Herrmann et al., 2015). The values of neighboring parameters in SAR for strained and non-strained rings remain somewhat ambiguous, again due to the very limited number of experimentally measured $k_{OH_{aq}}$ values for such molecules. Overall, the updated SAR (Model 2) showed good accuracy in the temperature range between 278-328K, which also underlines the need to further expand the kinetic databases to include a larger number of the temperature-dependent $k_{OH_{aq}}$ values.

The atmospheric lifetimes estimated for the five TAs - fenchol, borneol, menthol, camphanediol, and pinanediol – underlined the need to further investigate other non-acidic precursors of aqSOAs in the atmosphere. Recently, in connection with the formation of aqSOAs, a lot of attention has been dedicated to studying acidic precursors; such as terpenoic acids. Higher-MW organic acids are expected to reside almost entirely in the aqueous phase, even at relatively low values of LWC. Likewise, all TAs, and especially diols, studied in this work can undergo RI in clouds, fogs, and wet aerosols under realistic atmospheric

conditions, which underlines the need to study the formation of low-volatility products from such precursors.

**Appendices**

Kinetic database of the $k_{OH_{aq}}$ values for alcohols, diols, carboxylic acids, and carboxylate ions measured at 298K and in the temperature range between 278 and 328 K; this work and literature data (Witkowski, 2023).

**Credit author statement**

Bartłomiej Witkowski: Conceptualization, Supervision, Original Draft, Data Curation, Methodology, Writing - Review & Editing. Priyanka Jain: Investigation, Formal analysis, Methodology, Validation, Data Curation, Writing - Review & Editing. Beata Wileńska: Methodology, Resources. Tomasz Gierczak: Project administration, Supervision, Funding acquisition, Resources, Writing - Review & Editing.

## Declaration of competing interest

The authors have no competing interests to declare.

## Acknowledgments

This project was funded by the Polish National Science Centre: grant number 2018/31/B/ST10/01865. This study was carried out at the Biological and Chemical Research Centre, the University of Warsaw, established within the project co-financed by the European Union from the European Regional Development Fund under the Operational Programme Innovative Economy, 2007 – 2013. We thank the anonymous reviewers for their insightful comments and suggestions that helped to enhance the scientific quality of this article.

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
