# Peer review of "Electronic Supplementary Material"

_EGUsphere, 2023_

## Referee Comment (RC1)

**Review of ACP submission 'Temperature-dependent aqueous OH kinetics of C2-C10 linear and terpenoid alcohols and diols: new rate coefficients, structure-activity relationship and atmospheric lifetimes' by B. Witkowski et al., MS egusphere-2023-1381**

General

This is a quite interesting study of the kinetics of OH radicals with a number of linear and branched, terpenoid alcohols in aqueous solution. This topic has been treated before quite extensively but the work contributes some new kinetic data of interest for atmospheric multiphase and aqueous chemistry.

The whole paper could benefit from a thorough revision. All Figures and data presented should be discussed in detail, I think there is more to discover in the big data set obtained here. At times, I am missing text describing more of the findings.

Details

Table 1: The literature coverage is not really complete. The reactions of alcohols with OH have been reviewed by the IUPAC task group on atmospheric chemical kinetics and, accordingly, an overview on OH + EtOH in aqueous solution can be found here = https://iupac-aeris.ipsl.fr/datasheets/pdf/AQ_OH_2.pdf . Checking this might be useful for other alcohols. If we stay with OH + EtOH, why are only three previous studies reported ? Is the IUPAC task group site is used, please reference it accordingly.

The errors given in Table 1 appear very small to me. How were they derived ? Some of the means values are given with too many figures in view of the derived mean error. Please check the results' numerical format after re-checking the error margins of the derived mean rate constants.

Figure 5: Can you do a plot of both the gas phase and the aqueous phase data in one plot, maybe you can add it as a panel ( C) ?  Are the rate constants about the same ? Are there differences ? Why ?

Table 2: How do you interpret the obtained values for $\Delta S^{\ddagger}$ on a molecular basis ? Check your sign for double-degga.

Line 376: This number is disputable. Cloud processing occurs for shorter periods interrupted by cloud evaporations. To which reference time do these 18 h refer ? The total lifetime of an aqueous particle ? That would be good to add.

---

## Author Comment (AC1)

We would like to thank the Editor and the anonymous referee for their comments and for considering our manuscript for publication in ACP after revisions. Our responses to the comments provided together with the changes made in the revised manuscript are provided below.

**Referee 2** *comments***:**

*Referee 2*

Could the authors explain the T-dep. SAR values in the manuscript text regarding the prediction of T-dep. OH radicals in aqueous solution, not just in the Supporting Information? For example, by comparing the SAR rate constants to the measurements that were not part of the training set in an Arrhenius plot.

**Author's response:** We agree with this comment, the SAR performance at different temperatures was not sufficiently discussed. Please note however that we did not separate the compiled kinetic dataset into the training and evaluation sets. We believe that such an approach is adequate only when larger datasets are available to train the predictive model. At the same time, the current the aqueous kinetic databases still contain a relatively low number of rate coefficients, especially when compared with gas-kinetic databases.

**Changes in the manuscript:** The original Table S6 was removed from the SI. Figs. 10 and 11 were added in the main text to illustrate the performance at different temperatures. Table S9 was added in the SI, listing the results of the linear regression analysis of the measured (independent variable) vs predicted (dependent variable) $k_{OH}$ for the compounds used to optimize SAR factors at different temperatures – this data are presented in Fig. 11 and discussed in the main text – lines 460-464 and 473-480.

*Referee 2*

Authors should still check the revised manuscript and Supporting Information for spacing errors and typographical errors, such as in Table S3.

**Author's response:** We thank you for this comment.

**Changes in the manuscript:** The entire manuscript was revised (please see also the annotated version of the revised manuscript) and the spacing errors in the SI were corrected, whenever possible, and the volume of the SI was reduced.

---

## Author Comment (AC2)

We would like to thank the Editor and the anonymous referee for their comments and for considering our manuscript for publication in ACP after revisions. Our responses to the comments provided together with the changes made in the revised manuscript are provided below.

**Referee 1 comments*:**

**Referee 1*:**

The whole paper could benefit from a thorough revision. All Figures and data presented should be discussed in detail, I think there is more to discover in the big data set obtained here. At times, I am missing text describing more of the findings.

**Author's response:** We agree with these comments

**Changes in the manuscript:** The entire manuscript was revised (please see also the annotated manuscript), and the data obtained in the work presented (and the relevant literature data) are now discussed in much more detail. These revisions have led to new conclusions regarding the interplay between solvent effects (hydrophobic interactions and H-bonds) and the measured values of $k_{OH}$ for AAs studied in this work – lines 269-314 in the revised manuscript.

**Referee 1*:**

**Table 1:** The literature coverage is not really complete. The reactions of alcohols with OH have been reviewed by the IUPAC task group on atmospheric chemical kinetics.

**Author's response:** We agree with this comment, the data compiled by the IUPAC task group should have been referenced in the manuscript.

**Changes in the manuscript:** The recommended $k_{OH}$ values data and data included in the IUPAC datasheets were added in Table 1. The IUPAC task group website is now referenced in the revised manuscript. The discussion (lines 228-245 in the revised manuscript.) and footnotes connected with Table 1 were expanded and revised accordingly. Figures 4 and 5 were also revised, to include data listed in the IUPAC task group evaluations and the recommended $k_{OH}$ values.

Also, to further investigate the discrepancies between the $k_{OH_{aq}}$ values measured in this work and the recommended values, additional measurements were carried out for 1-propanol and 1 and 2-butanols using ethanol as the kinetic reference compounds - Table S8 was added to the SI. Discussion and captions connected with Figs. 4 and 5 were also revised after including the recommended IUPAC values. Figs. 7 and 8 were also modified to include the recommended $k_{OH}$ values included in the IUPAC task group datasheets, when available. The recommended values were included in the revised Figs. 7 and 8, and the discussion connected with these figures was revised accordingly – lines 368-378 and 384-393 in the revised manuscript.

*Referee 1*:

The errors given in Table 1 appear very small to me. How were they derived ? Some of the means values are given with too many figures in view of the derived mean error. Please check the results numerical format after re-checking the error margins of the derived mean rate constants.

**Author's response:** The experimental uncertainties given in Table 1 were calculated as two values of standard deviation from three or more separate measurements (Please see also section 2.6), which reflected only the precision of our measurements. We agree that this approach often resulted in unrealistically low uncertainties for some of the measured $k_{OH_{aq}}$ values.

**Changes in the manuscript:** The derivation of errors was revised, to include both the precision of our measurements and the uncertainties of the $k_{ref}$ values reported in the literature - see eq. VI in the main text. Section S6 was added to the SI, for a derivation of this formula – eq. SV and SVI. This new uncertainty analysis was done for every temperature - following this change, Table S6 in the SI was updated. The $k_{OH}$ values measured were rounded up to two significant figures, according to the $k_{ref}$ values used in the relative kinetic measurements. We believe that such an approach resulted in a much more realistic estimate of the errors for the $k_{OH_{aq}}$ values measured in the work presented.

*Referee 1*:

Figure 5: Can you do a plot of both the gas phase and the aqueous phase data in one plot, maybe you can add it as a panel (C)? Are the rate constants about the same? Are there differences ? Why?

**Author's response:** We agree with this comment.

**Changes in the manuscript** Fig. 5 was revised; panel C was added with the comparison of $k_{OH_{gas}}$ and $k_{OH_{aq}}$ values (in the same units) for homolog series of n-alkanes, n-alcohols and α,ω-diols. The data presented in Fig. 5C is now discussed in the revised manuscript – lines 295-309 in the revised manuscript.

*Referee 1*:

Table 2: How do you interpret the obtained values for S on a molecular basis? Check your sign for double-degga.

**Author's response:** We agree with this comment, the values of the activation parameters were not sufficiently discussed.

**Changes in the manuscript:** The typographic errors, including the double-degga sign pointed out by the referee, were corrected. More detailed discussion was added, in connection with the values of activation parameters obtained for the AAs under investigation. The following interpretation of the $\Delta S^{\ddagger}$ was added (lines 339-345):

"Negative values of $\Delta S^{\ddagger}$ were obtained for all AAs under investigation (Table 2). Higher (less negative) values of $\Delta S^{\ddagger}$ were obtained for the higher-MW AAs (including also cyclic AAs – Table 2), corresponding to a smaller decrease in disorder following R(I). Such a result may be associated with a release of water molecules from the hydrophobic surfaces (here alkyl chains) (Kroflič et al., 2020). As previously concluded, for some phenols, this phenomenon led to positive (or close to zero) values of $\Delta S^{\ddagger}$, even though the formation of a new bond between the reactants leads to a decrease in the disorder (Kroflič et al., 2020). The increase in the values of $\Delta S^{\ddagger}$ for the higher-MW AAs (with longer hydrophobic alkyl chains) also supports the assumption that the HIs influence the mechanism of R(I)."

Furthermore, Fig. 6 was added in the revised manuscript, to further discuss the trends in the changes of the values of activation parameters (derived in this work and literature values) in the homolog series of $C_1$-$C_{10}$, linear n-alcohols, and $\alpha,\omega$-diols. Fig. 6 includes the data obtained in the work presented as well as literature data compiled by the IUPAC task group. A paragraph with the discussion connected with Fig. 6 was added in the revised manuscript – lines 327 – 353.

*Referee 1*:

Line 376: This number is disputable. Cloud processing occurs for shorter periods interrupted by cloud evaporations. To which reference time do these 18 h refer ? The total lifetime of an aqueous particle ? That would be good to add.

**Author's response:** We agree with this comment; the time scales of cloud-water processing of WSOCs were not sufficiently discussed.

**Changes in the manuscript:** This paragraph was expanded, to include a more detailed discussion about clouds' interactions with air masses and the lifetimes of individual droplets in the context of the lifetimes shown in Fig. 12 (formerly Fig. 9) - lines 502-506:

"When this simplified model (eq. VII) is used, the time scales of interactions of gaseous species (air masses) also have to be considered. An average, weighted residence time of an air parcel inside a cloud was estimated as approx. 3h; by taking into account the total lifetime of a typical air parcel (~ 5 days), it will spend a total of 18 h inside a cloud (Herrmann et al., 2015). Furthermore, the aqueous-phase processing of organic molecules (here TAs) in the atmosphere is interrupted by the evaporation of cloud and fog droplets, which occurs on a time scale of a few minutes (Ervens, 2015)."

**Cited references:**

Ervens, B.: Modeling the Processing of Aerosol and Trace Gases in Clouds and Fogs, 115, 4157-4198, https://doi.org/10.1021/cr5005887, 2015.

Herrmann, H., Schaefer, T., Tilgner, A., Styler, S. A., Weller, C., Teich, M., and Otto, T.: Tropospheric Aqueous-Phase Chemistry: Kinetics, Mechanisms, and Its Coupling to a Changing Gas Phase, Chem. Rev., 115, 4259-4334, http://dx.doi.org/10.1021/cr500447k, 2015.

Kroflič, A., Schaefer, T., Huš, M., Phuoc Le, H., Otto, T., and Herrmann, H.: OH radicals reactivity towards phenol-related pollutants in water: temperature dependence of the rate constants and novel insights into the [OH–phenol]· adduct formation, 22, 1324-1332, https://doi.org/10.1039/C9CP05533A, 2020.